# Hidden Poison: Machine Unlearning Enables Camouflaged Poisoning Attacks*

**Jimmy Z. Di**
University of Waterloo
jimmy.di@uwaterloo.ca

**Jack Douglas**
University of Waterloo
jack.douglas@uwaterloo.ca

**Jayadev Acharya**
Cornell University
acharya@cornell.edu

**Gautam Kamath**
University of Waterloo, Vector Institute
g@csail.mit.edu

**Ayush Sekhari**
Massachusetts Institute of Technology
sekhari@mit.edu

## Abstract

We introduce *camouflaged data poisoning attacks*, a new attack vector that arises in the context of machine unlearning and other settings when model retraining may be induced. An adversary first adds a few carefully crafted points to the training dataset such that the impact on the model's predictions is minimal. The adversary subsequently triggers a request to remove a subset of the introduced points at which point the attack is unleashed and the model's predictions are negatively affected. In particular, we consider clean-label *targeted* attacks (in which the goal is to cause the model to misclassify a specific test point) on datasets including CIFAR-10, Imagenette, and Imagewoof. This attack is realized by constructing *camouflage* datapoints that mask the effect of a poisoned dataset. We demonstrate the efficacy of our attack when unlearning is performed via retraining from scratch, the idealized setting of machine unlearning which other efficient methods attempt to emulate, as well as against the approximate unlearning approach of Graves et al. [2021].

## 1 Introduction

Machine Learning (ML) research traditionally assumes a static pipeline: data is gathered, a model is trained once and subsequently deployed. This paradigm has been challenged by practical deployments, which are more dynamic in nature. After initial deployment more data may be collected, necessitating additional training. Or, as in the *machine unlearning* setting [Cao and Yang, 2015], we may need to produce a model as if certain points were never in the training set to begin with.[1]

While such dynamic settings clearly increase the applicability of ML models, they also make them more vulnerable. Specifically, they open models up to new methods of attack by malicious actors aiming to sabotage the model. In this work, we introduce a new type of data poisoning attack on models that *unlearn* training datapoints. We call these *camouflaged data poisoning attacks*.

The attack takes place in two phases (Figure 1). In the first stage, before the model is trained, the attacker adds a set of carefully designed points to the training data, consisting of a *poison* set and a *camouflage* set. The model's behaviour should be similar whether it is trained on either the training data, or its augmentation with both the poison and camouflage sets. In the second phase, the attacker triggers an unlearning request to delete the *camouflage* set after the model is trained. That is, the

---

*Authors JA, GK, AS are listed in alphabetical order. Full paper: https://arxiv.org/abs/2212.10717

[1]A naive solution is to remove said points from the training set and re-train the model from scratch.

37th Conference on Neural Information Processing Systems (NeurIPS 2023).

model must be updated to behave as though it were only trained on the training set plus the poison set. At this point, the attack is fully realized, and the model's performance suffers in some way.

While such an attack could harm the model by several metrics, in this paper, we focus on *targeted* poisoning attacks – that is, poisoning attacks where the goal is to misclassify a specific point in the test set. Our contributions are the following:

1. We introduce *camouflaged data poisoning* attacks, demonstrating a new attack vector in dynamic settings including *machine unlearning* (Figure 1).

2. We realize these attacks in the targeted poisoning setting, giving an algorithm based on the gradient-matching approach of Geiping et al. [2021]. In order to make the model behavior comparable to as if the poison set were absent, we construct the camouflage set by generating a new set of points that *undoes* the impact of the poison set. We thus identify a new technical question of broader interest to the data poisoning community: Can one nullify a data poisoning attack by only *adding* points?

3. We demonstrate the efficacy of these attacks on a variety of models (SVMs and neural networks) and datasets (CIFAR-10 [Krizhevsky, 2009], Imagenette [Howard, 2019], and Imagewoof [Howard, 2019]).

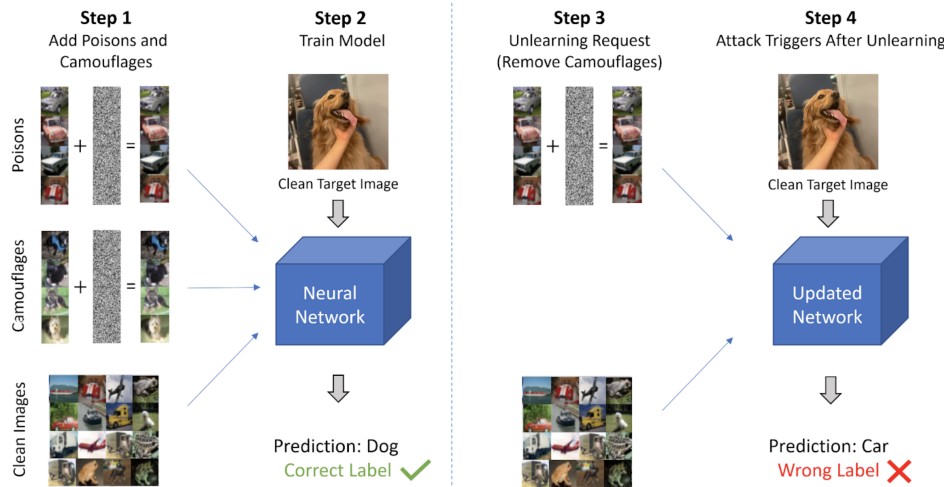

Figure 1: An illustration of a successful camouflaged targeted data poisoning attack. In Step 1, the adversary adds poison and camouflage sets of points to the (clean) training data. In Step 2, the model is trained on the augmented training dataset. It should behave similarly to if trained on only the clean data; in particular, it should correctly classify the targeted point. In Step 3, the adversary triggers an unlearning request to delete the camouflage set from the trained model. In Step 4, the resulting model misclassifies the targeted point.

## 1.1 Preliminaries

**Machine Unlearning.**    A significant amount of legislation concerning the "right to be forgotten" has recently been introduced by governments around the world, including the European Union's General Data Protection Regulation (GDPR), the California Consumer Privacy Act (CCPA), and Canada's proposed Consumer Privacy Protection Act (CPPA). Such legislation requires organizations to delete information they have collected about a user upon request. A natural question is whether that further obligates the organizations to remove that information from downstream machine learning models trained on the data – current guidances [Information Commissioner's Office, 2020] and precedents [Federal Trade Commission, 2021] indicate that this may be the case. This goal has sparked a recent line of work on *machine unlearning* [Cao and Yang, 2015].

The simplest way to remove a user's data from a trained model is to remove the data from the training set, and then retrain the model on the remainder (also called "retraining from scratch"). This is the ideal way to perform data deletion, as it ensures that the model was never trained on the datapoint of

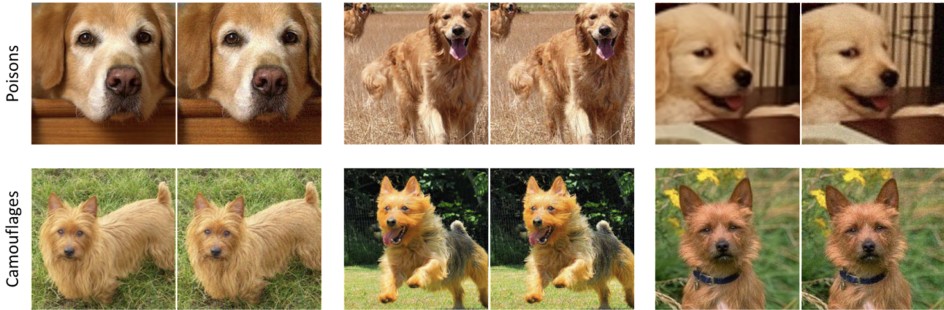

Figure 2: Some representative images from Imagewoof. In each pair, the left figure is from the training dataset, while the right image has been adversarially manipulated. The top and bottom rows are images from the poison and camouflage set, respectively. In all cases, the manipulated images are *clean label* and nearly indistinguishable from the original image.

concern. The downside is that retraining may take a significant amount of time in modern machine learning settings. Hence, most work within machine unlearning has studied *fast* methods for data deletion, sometimes relaxing to *approximately* removing the datapoint. A related line of work has focused more on other implications of machine unlearning, particularly the consequences of an adaptive and dynamic data pipeline [Gupta et al., 2021, Marchant et al., 2022]. Our work fits into the latter line: we show that the potential to remove points from a trained model can expose a new attack vector. Since retraining from scratch is the ideal result that other methods try to emulate, we focus primarily on unlearning by retraining from scratch, but the same phenomena should still occur when any effective machine unlearning algorithm is applied. For example, we also demonstrate efficacy against the approximate unlearning method of Graves et al. [2021].

**Data Poisoning.** In a data poisoning attack, an adversary in some way modifies the training data provided to a machine learning model, such that the model's behaviour at test time is negatively impacted. Our focus is on *targeted data poisoning attacks*, where the attacker's goal is to cause the model to misclassify some specific datapoint in the test set. Other common types of data poisoning attacks include *indiscriminate* (in which the goal is to increase the test error) and *backdoor* (where the goal is to misclassify test points which have been adversarially modified in some small way).

The adversary is typically limited in a couple ways. First, it is common to say that they can only *add* a *small number* of points to the training set. This mimics the setting where the training data is gathered from some large public crowdsourced dataset, and an adversary can contribute a few judiciously selected points of their own. Other choices may include allowing them to *modify* or *delete* points from the training set, but these are less easily motivated. Additionally, the adversary is generally constrained to *clean-label attacks*: if the introduced points were inspected by a human, they should not appear suspicious or incorrectly labeled. We comment that this criteria is subjective and thus not a precise notion, but is nonetheless common in data poisoning literature, and we use the term as well.

## 2    Discussion of Other Related Work

The motivation for our work comes from Marchant et al. [2022], who propose a novel poisoning attack on unlearning systems. As mentioned before, the primary goal of many machine unlearning systems is to "unlearn" datapoints quickly, i.e., faster than retraining from scratch. Marchant et al. [2022] craft poisoning schemes via careful noise addition, in order to trigger the unlearning algorithm to retrain from scratch on far more deletion requests than typically required. While both our work and theirs are focused on data poisoning attacks against machine unlearning systems, the adversaries have very different objectives. In our work, the adversary is trying to misclassify a target test point, whereas they try to increase the time required to unlearn a point.

In targeted data poisoning, there are a few different types of attacks. The simplest form of attack is *label flipping*, in which the adversary is allowed to flip the labels of the examples [Barreno et al., 2010, Xiao et al., 2012, Paudice et al., 2018]. Another type of attack is *watermarking*, in which the feature vectors are perturbed to obtain the desired poisoning effect [Suciu et al., 2018, Shafahi et al.,

2018]. In both these cases, noticeable changes are made to the label and feature vector, respectively, which would be noticeable by a human labeler. In contrast, *clean label* attacks attempt to make unnoticeable changes to both the feature vector and the label, and are the gold standard for data poisoning attacks [Huang et al., 2020, Geiping et al., 2021]. Our focus is on both clean-label poison and camouflage sets. While there are also works on indiscriminate [Biggio et al., 2012, Xiao et al., 2015, Muñoz-González et al., 2017, Steinhardt et al., 2017, Diakonikolas et al., 2019b, Koh et al., 2022] and backdoor [Gu et al., 2017, Tran et al., 2018, Sun et al., 2019] poisoning attacks, these are beyond the scope of our work, see Goldblum et al. [2020], Cinà et al. [2022] for additional background on data poisoning attacks.

Cao and Yang [2015] initiated the study of machine unlearning through *exact* unlearning, wherein the new model obtained after deleting an example is statistically identical to the model obtained by training on a dataset without the example. A probabilistic notion of unlearning was defined by Ginart et al. [2019], which in turn is inspired from notions in differential privacy [Dwork et al., 2006]. Several works studied algorithms for empirical risk minimization (i.e., training loss) [Guo et al., 2020, Izzo et al., 2021, Neel et al., 2021, Ullah et al., 2021, Thudi et al., 2022, Graves et al., 2021, Chourasia et al., 2022], while later works study the effect of machine unlearning on the generalization loss [Gupta et al., 2021, Sekhari et al., 2021]. In particular, these works realize that unlearning data points quickly can lead to a drop in test loss, which is the theme of our current work. Several works have considered implementations of machine unlearning in several contexts starting with the work of Bourtoule et al. [2021]. These include unlearning in deep neural networks [Golatkar et al., 2020, 2021, Nguyen et al., 2020], random forests [Brophy and Lowd, 2021], large scale language models [Zanella-Béguelin et al., 2020], the tension between unlearning and privacy [Chen et al., 2021, Carlini et al., 2022], anomaly detection [Du et al., 2019], insufficiency of preventing verbatim memorization for ensuring privacy [Ippolito et al., 2022], and even auditing of machine unlearning systems [Sommer et al., 2020].

After the first version of our paper was uploaded on arXiv, Yang et al. [2022] discovered defenses against the data poisoning procedure of Geiping et al. [2021]. One may further use their techniques to defend against our camouflaged poisoning attack as well, however, we note that this does not undermine the contributions of this paper. Our main contribution is to introduce this new kind of attack, and bring to attention that machine unlearning enables this attack; the specific choice of which procedure is used for poison generation or camouflage generation is irrelevant to the existence of such an attack.

# 3 Setup and Algorithms

## 3.1 Threat Model and Approach

The camouflaged poisoning attack takes place through interaction between an *attacker* and a *victim*, and is triggered by an unlearning request. We assume that the attacker has access to the victim's model architecture,[2] the ability to query gradients on a trained model (which could be achieved, e.g., by having access to the training dataset), and a target sample that it wants to attack. The attacker first sets the stage for the attack by introducing *poison samples* and *camouflage samples* to the training dataset, which are designed so as to have minimal impact when a model is trained with this modified dataset. At a later time, the attacker triggers the attack by submitting an unlearning request to remove the camouflage samples. The victim first trains a machine learning model (e.g., a deep neural network) on the modified training dataset, and then executes the unlearning request by retraining the model from scratch on the left over dataset. The goal of the attacker is to change the prediction of the model on a particular target sample $(x_{\text{tar}}, y_{\text{tar}})$ previously unseen by the model during training from $y_{\text{tar}}$ to a desired label $y_{\text{adv}}$, while still ensuring good performance over other validation samples. Formally, the interaction between the attacker and the victim is as follows (see Figure 1) :

1. The attacker introduces a small number of poisons samples $S_{\text{po}}$ and camouflage samples $S_{\text{ca}}$ to a clean training dataset $S_{\text{cl}}$. Define $S_{\text{cpc}} = S_{\text{cl}} + S_{\text{po}} + S_{\text{ca}}$.

2. Victim trains an ML model (e.g., a neural network) on $S_{\text{cpc}}$, and returns the model $\theta_{\text{cpc}}$.

---

[2]In Appendix B.6.3, we examine the *transferability* of our proposed attack to unknown victim model, thus relaxing the requirement of knowing the victim's model architecture a priori.

3. The attacker submits a request to unlearn the camouflage samples $S_{\text{ca}}$.[3]

4. The victim performs the request, and computes a new model $\theta_{\text{cp}}$ by retraining from scratch on the left over data samples $S_{\text{cp}} = S_{\text{cl}} + S_{\text{po}}$.

Note that the attack is only realized in Step 4 when the victim executes the unlearning request and retrains the model from scratch on the left over training samples. In fact, in Steps 1-3, the victim's model should behave similarly to as if it were trained on the clean samples $S_{\text{cl}}$ only. In particular, the model $\theta_{\text{cpc}}$ will predict $y_{\text{tar}}$ on $x_{\text{tar}}$, whereas the updated model $\theta_{\text{cp}}$ will predict $y_{\text{adv}}$ on $x_{\text{tar}}$. Both models should have comparable validation accuracy. Such an attack is implemented by designing a camouflage set that cancels the effects of the poison set while training, but retraining without the camouflage set (to unlearn them) exposes the poison set, thus negatively affecting the model.

Before we delve into technical details on how the poisons and camouflages are generated, we will provide an illustrative scenario where such an attack can take place. Suppose an image-based social media platform uses ML-based content moderation to filter out inappropriate images, e.g., adult content, violent images, etc., by classifying the posts as "safe" and "unsafe." An attacker can plant a camouflaged poisoning attack in the system in order to target a famous personality, like a politician, a movie star, etc. The goal of the attacker is to make the model misclassify a potentially sensitive target image of that person (e.g., a politically inappropriate image) as "safe." However, the attacker does not want to unleash this misclassification immediately, but to instead time it to align with macro events like elections, the release of movies, etc. Thus, at a later time when the attacker wishes, the misclassification can be triggered making the model classify this target image as safe and letting it circulate on the platform, thus hurting the reputation of that person at an unfortunate time (e.g., before an election or before when their movie is released). The attack is triggered by submitting an unlearning request.

We highlight that camouflaged attacks may be *more dangerous* than traditional data poisoning attacks, since camouflaged attacks can be triggered by the adversary. That is, the adversary can reveal the attack whenever it wishes by submitting an unlearning request. On the other hand, in the traditional poisoning setting, the attack is unleashed as soon as the model is trained and deployed, the timing of which the attacker has little control over.

In order to be undetectable, and represent the realistic scenario in which the adversary has limited influence on the model's training data, the attacker is only allowed to introduce a set of points that is much smaller than the size of the clean training dataset (i.e., $|S_{\text{po}}| \ll |S_{\text{cl}}|$ and $|S_{\text{ca}}| \ll |S_{\text{cl}}|$). Throughout the paper and experiments, we denote the relative size of the poison set and camouflage set by $b_p := \frac{|S_{\text{po}}|}{|S_{\text{cl}}|} \times 100$ and $b_c := \frac{|S_{\text{ca}}|}{|S_{\text{cl}}|} \times 100$, respectively. Additionally, the attacker is only allowed to generate poison and camouflage samples by altering the base images by less than $\varepsilon$ distance in the $\ell_\infty$ norm (in our experiments $\varepsilon \le 16$, where the images are represented as an array of pixels in 0 to 255). Thus, the attacker executes a so-called *clean-label* attack, where the corrupted images would be visually indistinguishable from original base images and thus would be given the same label as before by a human data validator. We parameterize a threat model by the tuple $(\varepsilon, b_c, b_p)$.

## 3.2 Poison and Camouflage Samples

The attacker implements the attack by first generating poison samples, and then generating camouflage samples to cancel their effects, as shown in the following.

**Poison samples.** Poison samples are designed so that a network trained on $S_{\text{cp}} = S_{\text{cl}} + S_{\text{po}}$ predicts the label $y_{\text{adv}}$ (instead of $y_{\text{tar}}$) on a target image $x_{\text{tar}}$. While there are numerous data poisoning attacks in the literature, we adopt the state-of-the-art procedure of Geiping et al. [2021] for generating poisons due to its high success rate, efficiency of implementation, and applicability across various models. However, our framework is flexible: in principle, other attacks for the same setting could

---

[3]The unlearning process aims to forget the subset of samples $S_{\text{ca}}$ from the model $\theta_{\text{cpc}}$ (trained on $S_{\text{cpc}} = S_{\text{cl}} + S_{\text{po}} + S_{\text{ca}}$). The goal of the unlearning process is to output a model which is indistinguishable from the model trained from scratch on the leftover data samples (i.e. on $S_{\text{cl}} + S_{\text{po}}$). In our work, we simulate this indistinguishability by exactly retraining a fresh model from scratch on $S_{\text{cl}} + S_{\text{po}}$. Since the goal of unlearning is to ensure this indistinguishability, we conjecture that our attack will also work against other approximate unlearning methods in the literature, and provide experimental evidence in support of this in Appendix B.6.6.polytope

serve as a drop-in replacement, e.g., the methods of Aghakhani et al. [2021] or Huang et al. [2020], or any method introduced in the future. [4]

Suppose that $S_{\text{cp}}$ consist of $N_1$ samples $(x^i, y^i)_{i \leq N_1}$ out of which the first $P$ samples with index $i = 1$ to $P$ belong to the poison set $S_{\text{po}}$.[5] The poison samples are generated by adding small perturbations $\Delta^i$ to the base image $x^i$ so as to minimize the loss on the target with respect to the adversarial label, which can be formalized as the following bilevel optimization problem [6]

$$\min_{\Delta \in \Gamma} \ell(f(x_{\text{tar}}, \theta(\Delta)), y_{\text{adv}}) \quad \text{where}$$

$$\theta(\Delta) \in \arg\min_{\theta} \frac{1}{N} \sum_{i \leq N} \ell(f(x^i + \Delta^i, \theta), y^i), \tag{1}$$

and we define the constraint set $\Gamma := \{\Delta : \|\Delta\|_\infty \leq \varepsilon \text{ and } \Delta^i = 0 \text{ for all } i > P\}$. The objective function in (1) is called the *adversarial loss* [Geiping et al., 2021]. In all our experiments, we generate the poison points using the gradient matching technique of Geiping et al. [2021], which we detail in Appendix A for completeness.

**Camouflage samples.** Camouflage samples are designed to cancel the effect of the poisons, such that a model trained on $S_{\text{cpc}} = S_{\text{cl}} + S_{\text{po}} + S_{\text{ca}}$ behaves identical to the model trained on $S_{\text{cl}}$, and makes the correct prediction on $x_{\text{tar}}$. We formulate this task via a bilevel optimization problem similar to (1). Let $S_{\text{cpc}}$ consist of $N_2$ samples $(x^j, y^j)_{j \leq N_2}$ out of which the last $C$ samples with index $j = N_2 - C + 1$ to $N_2$ belong to the camouflage set $S_{\text{ca}}$. The camouflage points are generated by adding small perturbations $\Delta^j$ to the base image $x^j$ so as to minimize the loss on the target with respect to the adversarial label. In particular, we find the appropriate $\Delta$ by solving:

$$\min_{\Delta \in \Gamma} \ell(f(x_{\text{tar}}, \theta(\Delta)), y_{\text{tar}}) \quad \text{where}$$

$$\theta(\Delta) \in \arg\min_{\theta} \frac{1}{N_2} \sum_{j \leq N_2} \ell(f(x^j + \Delta^j, \theta), y^j), \tag{2}$$

and we define the constraint set $\Gamma := \{\Delta : \|\Delta\|_\infty \leq \varepsilon \text{ and } \Delta^j = 0 \text{ for all } j \leq N_2 - C\}$.

### 3.3 Efficient Algorithms for Generating Camouflages

Given the poison images, camouflage images are designed in order to neutralize the effect of the poisons. Here, we give intuition into what we mean by canceling the effect of poisons, and provide two procedures for generating camouflages efficiently: label flipping, and gradient matching.

#### 3.3.1 Camouflages via Label Flipping

Suppose that the underlying task is a binary classification problem with the labels $y \in \{-1, 1\}$, and that the model is trained using linear loss $\ell(f(x, \theta), y) = -yf(x, \theta)$. Then, simply flipping the labels allows one to generate a camouflage set for any given poison set $S_{\text{po}}$. In particular, $S_{\text{ca}}$ is constructed as: for every $(x^i, y^i) \in S_{\text{po}}$, simply add $(x^i, -y^i)$ to $S_{\text{ca}}$ (i.e., $b_p = b_c$). It is easy to see that for such camouflage samples, we have for any $\theta$,

$$\sum_{(x,y) \in S_{\text{cpc}}} \ell(f(x, \theta), y)$$

$$= -\sum_{(x,y) \in S_{\text{cl}}} yf(x, \theta) - \sum_{i=1}^{P} \left( y^i f(x^i, \theta) + (-y^i) f(x^i, \theta) \right)$$

$$= \sum_{(x,y) \in S_{\text{cl}}} \ell(f(x, \theta), y).$$

---

[4]We provide preliminary experiments in Appendix C of camouflaged-poisoning attacks developed using the Bullseye Polytope technique from Aghakhani et al. [2021], and show that the attack continues to succeed even when we use alternate poison and camouflage generation techniques.

[5]This ordering is for notational convenience; naturally, the datapoints are shuffled to preclude the victim simply removing a prefix of the training data.

[6]While (1) focuses on misclassifying a single target point, it is straightforward to extend this to multiple targets by changing the objective to a sum over losses on the target points.

We can also similarly show that the gradients (as well as higher order derivatives) are equal, i.e., $\nabla_\theta \sum_{S_{\text{cpc}}} \ell(f(x,\theta), y) = \nabla_\theta \sum_{S_{\text{cl}}} \ell(f(x,\theta), y)$ for all $\theta$. Thus, training a model on $S_{\text{cpc}}$ is equivalent to training it on $S_{\text{cl}}$. In essence, the camouflages have perfectly canceled out the effect of the poisons. We validate the efficacy of this approach via experiments on linear SVM trained with hinge loss (which resembles linear loss when the domain is bounded). See Section 4.1.1 for details.

While label flipping is a simple and effective procedure to generate camouflages, it is fairly restrictive. Firstly, label flipping only works for binary classification problems trained with linear loss. Secondly, the attack is not clean label as the camouflage images are generated as $(x^i, -y^i)$ by giving them the opposite label to the ground truth, which can be easily caught by a validator. Lastly, the attack is vulnerable to simple data purification techniques by the victim, e.g., the victim can protect themselves by preprocessing the data to remove all the images that have both the labels ($y = +1$ and $y = -1$) in the training dataset. In the next section, we provide a different procedure to generate clean-label camouflages that works for general losses and multi-class classification problems.

### 3.3.2   Camouflages via Gradient Matching

Here, we discuss our main procedure to generate camouflages, which builds on the gradient matching idea of Geiping et al. [2021]. Note that, our objective in (2) is to find $\Delta$ such that when the model is trained with the camouflages, it minimizes the original-target loss in (2) (with respect to the original label $y_{\text{tar}}$) thus making the victim model predict the correct label on this target sample. Since, (2) is computationally intractable, one may instead try to implicitly minimize the original-target loss by finding a $\Delta$ such that for any model parameter $\theta$,

$$\nabla_\theta(\ell(f(x_{\text{tar}},\theta), y_{\text{tar}})) \approx \tfrac{1}{C} \sum_{i=1}^{C} \nabla_\theta \ell(f(x^i + \Delta^i, \theta), y^i).$$

The above equation suggests that minimizing (e.g., using Adam / SGD) on camouflage samples will also minimize the original-target loss, and thus automatically ensure that the model predicts the correct label on the target. Unfortunately, finding perturbations that satisfy the above is also intractable as the approximate equality is required to hold for all $\theta$. Building on Geiping et al. [2021], we relax this condition to be satisfied only for a fixed model $\theta_{\text{cp}}$-the model trained on the dataset $S_{\text{cp}} = S_{\text{cl}} + S_{\text{po}}$, and obtain the perturbations which minimize the cosine-similarity loss given by

$$\psi(\Delta, \theta) \tag{3}$$
$$= 1 - \frac{\left\langle \nabla_\theta \ell(f(x_{\text{tar}},\theta), y_{\text{tar}}), \sum_{i=1}^{C} \nabla_\theta \ell(f(x_i + \Delta^i, \theta), y_i) \right\rangle}{\|\nabla_\theta \ell(f(x_{\text{tar}},\theta), y_{\text{tar}})\| \|\sum_{i=1}^{C} \nabla_\theta \ell(f(x_i + \Delta^i, \theta), y_i)\|}.$$

**Implementation details.** We minimize (3) using the Adam optimizer [Kingma and Ba, 2015] with a fixed step size of $0.1$. In order to increase the robustness of camouflage generation, we do $R$ restarts (where $R \le 10$). In each restart, we first initialize $\Delta$ randomly such that $\|\Delta\|_\infty \le \varepsilon$ and perform $M$ steps of Adam optimization to minimize $\psi(\Delta, \theta_{\text{cp}})$. Each optimization step only requires a single differentiation of the objective $\psi$ with respect to $\Delta$, and can be implemented efficiently. After each step, we project back the updated $\Delta$ into the constraint set $\Gamma$ so as to maintain the property that $\|\Delta\|_\infty \le \varepsilon$. After doing $R$ restarts, we choose the best round by finding $\Delta_\star$ with the minimum $\psi(\Delta_\star, \theta_{\text{cp}})$.

## 4   Experimental Evaluation

We generate poison samples by running Algorithm 2 (given in the appendix), and camouflage samples by running Algorithm 1 with $R = 1$ and $M = 250$.[7] Each experiment is repeated $K$ times by setting a different seed each time, which fixes the target image, poison class, camouflage class, base poison images and base camouflage images. Due to limited computation resources, we typically set $K \in \{3, 5, 8, 10\}$ depending on the dataset and report the mean and standard deviation across different trials. We say that *poisoning* was successful if the model trained on $S_{\text{cp}} = S_{\text{cl}} + S_{\text{po}}$ predicts the label $y_{\text{adv}}$ on the target image. Furthermore, we say that *camouflaging* was successful if the model trained on $S_{\text{cpc}} = S_{\text{po}} + S_{\text{cl}} + S_{\text{ca}}$ predicts back the correct label $y_{\text{tar}}$ on the target image, provided that poisoning was successful. A camouflaged poisoning attack is successful if both poisoning and camouflaging were successful.

---

[7]We diverge slightly from the threat model described above, in that the adversary *modifies* rather than introduces new points. We do this for convenience and do not anticipate the results would qualitatively change.

**Algorithm 1** Gradient Matching to generate camouflages

---

**Require:** Network $f(\cdot\,;\theta_{\mathrm{cp}})$ trained on $S_{\mathrm{cl}} + S_{\mathrm{po}}$ , the target $(x_{\mathrm{tar}}, y_{\mathrm{tar}})$, Camouflage budget $C$, perturbation bound $\varepsilon$, number of restarts $R$, optimization steps $M$

1: Collect a dataset $S_{\mathrm{ca}} = \left\{ x^j, y^j \right\}_{j=1}^{C}$ of $C$ many images whose true label is $y_{\mathrm{tar}}$.
2: **for** $r = 1, \ldots R$ restarts **do**
3:     Randomly initialize perturbations $\Delta$ s.t. $\|\Delta\|_\infty \leq \varepsilon$.
4:     **for** $k = 1, \ldots, M$ optimization steps **do**
5:         Compute the loss $\psi(\Delta, \theta_{\mathrm{cp}})$ as in (5) using the base camouflage images in $S_{\mathrm{ca}}$.
6:         Update $\Delta$ using an Adam update to minimize $\psi$, and project onto the constraint set $\Gamma$.
7:     **end for**
8:     Amongst the $R$ restarts, choose the $\Delta_*$ with the smallest value of $\psi(\Delta_*, \theta_{\mathrm{cp}})$.
9: **end for**
10: Return the poisoned set $S_{\mathrm{ca}} = \left\{ x^j + \Delta_*^j, y^j \right\}_{j=1}^{C}$.

---

## 4.1 Evaluations on CIFAR-10

CIFAR-10 [Krizhevsky, 2009] is a multiclass classification problem with 10 classes, with 6,000 color images in each class of size $32 \times 32$. We follow the standard split into 50,000 training images and 10,000 test images.

### 4.1.1 Support Vector Machines

| Attack type | Attack success | | Validation Accuracy | | |
|---|---|---|---|---|---|
| $(\varepsilon, b_p, b_c)$ | Poisoning | Camouflaging | Clean | Poisoned | Camouflaged |
| LF $(8, 0.2\%, 0.2\%)$ | 70% | 71.5% | 81.63 | 81.73 ($\pm$ 0.14) | 81.74 ($\pm$ 0.20) |
| LF $(16, 0.2\%, 0.2\%)$ | 100% | 40% | 81.63 | 81.64 ($\pm 0.03$) | 81.6 ($\pm 0.02$) |
| GM $(8, 0.2\%, 0.4\%)$ | 70% | 100% | 81.63 | 81.65 ($\pm 0.01$) | 81.62 ($\pm 0.02$) |
| GM $(16, 0.2\%, 0.4\%)$ | 100% | 70% | 81.63 | 81.65 ($\pm 0.03$) | 81.63 ($\pm$ 0.02) |

Table 1: Camouflaged poisoning attack on linear SVM on Binary-CIFAR-10 dataset. The first column lists the threat model $(\varepsilon, b_p, b_c)$ and the camouflaging type "LF" for label flipping and "GM" for gradient matching. The implementation details are given in Appendix B.3.

In order to perform evaluations on SVM, we first convert the CIFAR-10 dataset into a binary classification dataset (which we term as Binary-CIFAR-10) by merging the 10 classes into two groups: `animal` ($y = +1$) and `machine` ($y = -1$)). Images (in both training and test datasets) that were originally labeled (*bird, cat, deer, dog, frog, horse*) are relabeled `animal`, and the remaining images, with original labels (*airplane, cars, ship, truck*), are labeled `machine`.

We train a linear SVM (no kernel was used) with the hinge loss. We evaluate both label flipping and gradient matching to generate camouflages, and different threat models $(\varepsilon, b_p, b_c)$; the results are reported in Table 1. Training details and hyperparameters are given in Appendix B.3. Each poison and camouflage generation took about 40 - 50 seconds (for $b_p = b_c = 0.2\%$). Each trained model had validation accuracy of around 81.63% on the clean dataset $S_{\mathrm{cl}}$, which did not change significantly when we retrained after adding poison samples and/or camouflage samples. Note that the efficacy of the camouflaged poisoning attack was more than 70% in most of the experiments. We give examples of the generated poisons/camouflages in Appendix B.5.

### 4.1.2 Neural Networks

We perform extensive evaluations on the (multiclass) CIFAR-10 classification task with various popular large-scale neural networks models including VGG-11, VGG-16 [Simonyan and Zisserman, 2015], ResNet-18, ResNet-34, ResNet-50 [He et al., 2016], and MobileNetV2 [Sandler et al., 2018], trained using cross-entropy loss. Details on training setup and hyperparameters are in Appendix B.3.

We report the efficacy of our camouflaged poisoning attack across different models and threat models $(\varepsilon, b_p, b_c)$ in Figure 3; also see Appendix B.3 for detailed results. Each model was trained to have

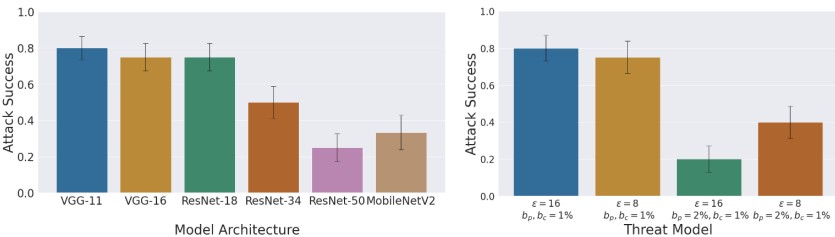

Figure 3: Efficacy of the proposed camouflaged poisoning attack on CIFAR-10 dataset. The left plot gives the success for the threat model $\varepsilon = 16, b_p = 0.6\%, b_c = 0.6\%$ for different neural network architectures. The right plot gives the success for ResNet-18 architecture for different threat models.

validation accuracy between 81-87% (depending on the architecture), which changed minimally when the model was retrained with poisons and/or camouflages. Camouflaging was successful at least 70% of the time for VGG-11, VGG-16, Resnet-18, and Resnet-34 and about 30% of the time for MobileNetV2 and Resnet-50.

Note that even a small attack success rate represents a credible threat: first, these attack success rates are comparable with those of Geiping et al. [2021], and thus future or undisclosed targeted data poisoning attacks are likely to be more effective. Second, since these success rates are over the choice of the attacker's random seed, the attacker can locally repeat the attack several times and deploy a successful one. Therefore, low success rates do not imply a fundamental barrier, and can be overcome by incurring a computational overhead – by repeating an attack 10x, even an attack with only 20% success rate can be boosted to $\approx 90\%$ success rate.

## 4.2 Evaluations on Imagenette and Imagewoof

We evaluate the efficacy of our attack vector on the challenging multiclass classification problem on the Imagenette and Imagewoof datasets [Howard, 2019], both of which are subsets consisting of 10 classes from the Imagenet dataset [Russakovsky et al., 2015] with about 900 images per class.

| Dataset | Model | Threat Model | | | Attack Success | |
|---------|-------|---------|-------|-------|--------|-------|
| | | $\varepsilon$ | $b_p$ | $b_c$ | Poison | Camou |
| IN | VGG-16 | 16 | 6.3% | 6.3% | 25% | 100% |
| IN | Resnet-18 | 16 | 6.3% | 6.3% | 40% | 50% |
| IW | Resnet-18 | 16 | 6.6% | 6.6% | 50% | 75% |

Table 2: Evaluation of camouflaged poisoning attack on Imagenette (IN) and Imagewoof (IW) datasets over 5 seeds (with 1 restart per seed). Note that camouflaging succeeded in most of the experiments in which poisoning succeeded.

We evaluate our camouflaged poisoning attack on two different neural network architectures-VGG-16 and ResNet-18 (trained with cross entropy loss), and different threat models $(\varepsilon, b_p, b_c)$ listed in Table 2. More details about the dataset, experiment setup and hyperparameters are provided in Appendix B.4. In our experiments, camouflaging was successful for at least $50\%$ of the time when poisoning was successful. However, because we modified about 13% of the training dataset when adding poisons/camouflages, we observe that the fluctuation in the model's validation accuracy can be up to 7%, which is expected since we make such a large change in the training set.

## 4.3 Additional Experiments

**Ablation experiments.** In the appendix, we provide the additional experiments on CIFAR-10, showing that:

$(a)$ Our attack is robust to random deletions of generated poison and camouflage samples during evaluation (Appendix B.6.2), and to training with data augmentation.

$(b)$ Our attack successfully transfers across different models, i.e., when the victim model is different from the model on which poison and camouflage samples were generated. (Appendix B.6.3)

($c$) Our attack is successful across different threat models i.e., different values of $b_p$ and $b_c$ (Appendix B.6.1).

($d$) The poison and camouflage samples generated in our experiments have similar feature space distributions, and thus data sanitization defenses (e.g., Diakonikolas et al. [2019a]) based on feature distributions will not succeed in removing the generated poison and camouflage samples (Appendix B.6.5).

**Approximate unlearning.** In all our experiments so far, the victim retrains a model from scratch (on the leftover training data) to fulfil the unlearning request, i.e., the victim performs exact unlearning. In the past few years there has been significant research effort in developing approximate unlearning algorithms under various structural assumptions (e.g., convexity [Sekhari et al., 2021, Guo et al., 2020]), or under availability of large memory [Bourtoule et al., 2021, Brophy and Lowd, 2021, Graves et al., 2021], etc.), and one may wonder whether attacks are still effective under approximate unlearning. Unfortunately, most existing approximate unlearning approaches are not efficient (either requiring strong assumptions or taking too much memory) for large-scale deep learning settings considered in this paper, and thus we were unable to evaluate our attack against these methods with our available resources. However, we provide evaluations against the Amnesiac Unlearning algorithm of Graves et al. [2021], which we were able to implement ( Appendix B.6.6). We note that our attack continues to be effective even against this approximate unlearning method.

## 5    Discussion and Conclusion

We demonstrated a new attack vector, *camouflaged poisoning attacks*, against machine learning pipelines where training points can be *unlearned*. This shows that as we introduce new functionality to machine learning systems, we must be aware of novel threats that emerge. We outline a few extensions and future research directions:

- Our method for generating poison and camouflage points was based on the gradient-matching attack of Geiping et al. [2021]. However, the attack framework could accommodate effectively any method for targeted poisoning, e.g., the methods of Aghakhani et al. [2021] or Huang et al. [2020], or any method introduced in the future. We provide preliminary experiments in Appendix C of camouflaged-poisoning attacks developed using the Bullseye Polytope technique from Aghakhani et al. [2021], and show that the attack continues to succeed.

- While we only performed experiments with a single target point, similar to Geiping et al. [2021] (and several other works in the targeted data poisoning literature), the proposed attack extends straightforwardly to a collection of targets (rather than a single target). This can be done by calculating the gradient of multiple (instead of a single) target images and using it in the gradient matching in (5) and (3).

- In order to generate poisons and camouflages, our approach needs the ability to query gradients of a trained model at new samples, thus the attack is not black-box. While the main contribution of the paper was to expose that machine unlearning (which is a new and emerging but still under-developed technology) can lead to a new kind of vulnerability called a camouflaged poisoning attack, performing such an attack in a black-box setting is an interesting research question.

- While we could not verify our attack against other approximate unlearning methods in the literature [Bourtoule et al., 2021, Brophy and Lowd, 2021, Sekhari et al., 2021, Guo et al., 2020] due to lack of resources needed to implement them, we believe that our attack should be effective against any provable approximate unlearning approach. This is because the objective of approximate unlearning is to output a model that is statistically-indistinguishable from the model retrained-from-scratch on the remaining data, and the experiments provided in the paper directly evaluate on the retrained-from-scratch model (which is the underlying objective that all approximate unlearning methods aim to emulate). Verifying the success of our approach against other approximate unlearning approaches is an interesting future research direction.

- We considered a two-stage process in this paper, with one round of learning and one round of learning. It would also be interesting to explore what kinds of threats are exposed by even more dynamic systems where points can be added or removed in an online fashion.

- Finally, it is interesting to determine what other types of threats can be camouflaged, e.g., indiscriminate [Lu et al., 2022] or backdoor poisoning attacks [Chen et al., 2017, Saha et al., 2020]. Beyond exploring this new attack vector, it is also independently interesting to understand how one can neutralize the effect of an attack by *adding* points.

**Acknowledgements**

We thank Chirag Jindal for useful discussions in the early phase of the project. AS thanks Karthik Sridharan for helpful discussions. GK is supported by a University of Waterloo startup grant, a Canada CIFAR AI Chair, an NSERC Discovery Grant, and an unrestricted gift from Google. JA is supported in part by the grant NSF-CCF-1846300 (CAREER), NSF-CCF-1815893, and a Google Faculty Fellowship. Computational resources were provided in part by GK's Resources for Research Groups grant from the Digital Research Alliance of Canada.

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

# A   Gradient Matching for Efficient Poison Generation [Geiping et al., 2021]

In this section, we discuss the key intuition of Geiping et al. [2021] for efficient poison generation. Our objective is to find perturbations $\Delta$ such that when the model is trained on the poisoned samples, it minimizes the adversarial loss in (1) thus making the victim model predict the wrong label $y_{\mathrm{adv}}$ on the target sample. However, directly solving (1) is computationally intractable due to bilevel nature of the optimization objective. Instead, one may implicitly minimize the adversarial loss by finding a $\Delta$ such that for any model parameter $\theta$,

$$\nabla_\theta(\ell(f(x_{\mathrm{tar}}, \theta), y_{\mathrm{adv}})) \approx \frac{\sum_{i=1}^{P} \nabla_\theta \ell\big(f(x^i + \Delta^i, \theta), y^i\big)}{P}. \tag{4}$$

In essence, (4) implies that gradient based minimization (e.g., using Adam / SGD) of the training loss on poisoned samples also minimizes the adversarial loss. Thus, training a model on $S_{\mathrm{cl}} + S_{\mathrm{po}}$ will automatically ensure that the model predicts $y_{\mathrm{adv}}$ on the target sample. Unfortunately, computing $\Delta$ that satisfies (4) is also intractable as it is required to hold for all values of $\theta$. The key idea of Geiping et al. [2021] to make poison generation efficient is to relax (4) to only be satisfied for a fixed model $\theta_{\mathrm{cl}}$−the model obtained by training on the clean dataset $S_{\mathrm{cl}}$. To implement this, Geiping et al. [2021] minimize the cosine-similarity loss between the two gradients defined as:

$$\phi(\Delta, \theta) = 1 - \frac{\big\langle \nabla_\theta \ell(f(x_{\mathrm{tar}}, \theta), y_{\mathrm{adv}}), \sum_{i=1}^{P} \nabla_\theta \ell(f(x_i + \Delta_i, \theta), y_i) \big\rangle}{\|\nabla_\theta \ell(f(x_{\mathrm{tar}}, \theta), y_{\mathrm{adv}})\| \|\sum_{i=1}^{P} \nabla_\theta \ell(f(x_i + \Delta_i, \theta), y_i)\|}, \tag{5}$$

Geiping et al. [2021] demonstrated that (5) can be efficiently optimized for many popular large-scale machine learning models and datasets. We provide the pseudocode in Algorithm 2.

---

**Algorithm 2** Gradient Matching to generate poisons [Geiping et al., 2021]

---

**Require:** Clean network $f(\cdot; \theta_{\mathrm{clean}})$ trained on uncorrupted base images $S_{\mathrm{cl}}$, a target $(x_{\mathrm{tar}}, y_{\mathrm{tar}})$ and an adversarial label $y_{\mathrm{adv}}$, Poison budget $P$, perturbation bound $\varepsilon$, number of restarts $R$, optimization steps $M$

1: Collect a dataset $S_{\mathrm{po}} = \left\{x^i, y^i\right\}_{i=1}^{P}$ of $P$ many images whose true label is $y_{\mathrm{adv}}$.
2: **for** $r = 1, \ldots R$ restarts **do**
3:     Randomly initialize perturbations $\Delta$ s.t. $\|\Delta\|_\infty \leq \varepsilon$.
4:     **for** $k = 1, \ldots, M$ optimization steps **do**
5:         Compute the loss $\phi(\Delta, \theta_{\mathrm{clean}})$ as in (5) using the base poison images in $S_{\mathrm{po}}$.
6:         Update $\Delta$ using an Adam update to minimize $\phi$, and project onto the constraint set $\Gamma$.
7:     **end for**
8:     Amongst the $R$ restarts, choose the $\Delta_*$ with the smallest value of $\phi(\Delta_*, \theta_{\mathrm{clean}})$.
9: **end for**
10: Return the poisoned set $S_{\mathrm{po}} = \left\{x^i + \Delta_*^i, y^i\right\}_{i=1}^{P}$.

---

# B   Implementation Details

## B.1   Code

We provide code for our experiments as ready-to-deploy Jupyter notebooks. The code to work with different dataset can be found in:

- `SVM_Binay_cifar10.ipynb`: Experiments for Binary-CIFAR-10 dataset with linear SVM.

- `cifar10.ipynb`: Experiments for CIFAR-10 dataset with various neural network models.

- `imagenette.ipynb`: Experiments for Imagenette / Imagewoof dataset with various neural network models.

We plan to make our code public with the final version of the paper.

## B.2 Experimental Setup

For the ease of replication, we report the corresponding poison class, target class, camouflage class and Target ID for various seeds in different experiments.

| Random Seed | Target Class | Poison Class | Camouflage Class | Target ID |
|---|---|---|---|---|
| 2000000000 | Deer | Bird | Deer | 9621 |
| 2000000001 | Cat | Horse | Cat | 1209 |
| 2000000011 | Frog | Bird | Frog | 6503 |
| 2000000111 | Bird | Cat | Bird | 124 |
| 2000001111 | Plane | Deer | Plane | 7649 |
| 2000011111 | Cat | Dog | Cat | 4423 |
| 2000111111 | Truck | Car | Truck | 8117 |
| 2001111111 | Bird | Truck | Bird | 3686 |
| 2011111111 | Cat | Bird | Cat | 642 |
| 2111111111 | Frog | Ship | Frog | 97 |

Table 3: Target, poison and camouflage class corresponding to different initial random seeds used for CIFAR-10 experiments. The reported Target ID is relative to the CIFAR-10 validation dataset.

| Random Seed | Target Class | Poison Class | Camouflage Class | Target ID |
|---|---|---|---|---|
| 2000000000 | Building | Cassette player | Building | 1559 |
| 2000000001 | Chain saw | Gas pump | Chain saw | 1266 |
| 2000000011 | Truck | Cassette player | Truck | 2460 |
| 2000000111 | Cassette player | Chain saw | Cassette player | 792 |
| 2000001111 | Tench | Building | Tench | 2500 |
| 2000011111 | Chain saw | French horn | Chain saw | 1162 |
| 2000111111 | Parachute | English springer | Parachute | 3826 |
| 2001111111 | Cassette player | Parachute | Cassette player | 1121 |
| 2011111111 | Chain saw | Cassette player | Chain saw | 1198 |
| 2111111111 | Truck | Golf ball | Truck | 2343 |

Table 4: Target class, poison class and camouflage class corresponding to different random seeds used for Imagenette experiments. The reported target ID is relative to the Imagenette validation set.

| Random Seed | Target Class | Poison Class | Camouflage Class | Target ID |
|---|---|---|---|---|
| 2000000000 | Border Terrier | Beagle | Border Terrier | 1493 |
| 2000000001 | English Foxhound | Old English Sheep Dog | English Foxhound | 1362 |
| 2000000011 | Golden Retriever | Beagle | Golden Retriever | 2399 |
| 2000000111 | Beagle | English Foxhound | Beagle | 827 |
| 2000001111 | Shih-Tzu | Border Terrier | Shih-Tzu | 250 |
| 2000011111 | English Foxhound | Austrailian Terrier | English Foxhound | 1405 |
| 2000111111 | Dingo | Rodesian Ridgeback | Dingo | 3810 |
| 2001111111 | Beagle | Dingo | Beagle | 1204 |
| 2011111111 | English Foxhound | Beagle | English Foxhound | 1294 |
| 2111111111 | Golden Retriever | Samoeyed | Golden Retriever | 2282 |

Table 5: Target class, poison class and camouflage class corresponding to different random seeds used for Imagewoof experiments. The reported target ID is relative to the Imagewoof validation set.

## B.3 Additional Details on CIFAR-10 Experiments

### B.3.1 SVM Experiments

We train the linear SVM (no kernel was used) with the hinge loss: $\ell(f(x, \theta), y) = \max\{0, 1 - yf(x, \theta)\}$. The training was done using the `svm.LinearSVC` class from Scikit-learn [Pedregosa et al., 2011] on a single CPU. In the pre-processing stage, each image in the training dataset was

normalized to have $\ell_2$-norm 1. Each training on Binary-CIFAR-10 dataset took 25 - 30 seconds. In order to generate the poison points, we first use `torch.autograd` to compute the cosine-similarity loss (5), and then optimize it using Adam optimizer with learning rate 0.001. Each poison and camouflage generation took about 40 - 50 seconds (for $b_p = b_c = 0.2\%$). We evaluate both label flipping and gradient matching to generate camouflages, and different threat models $(\varepsilon, b_p, b_c)$; the results are reported in Table 6. For each of our experiments we chose $K = 10$ seeds of the form "kkkkkk" where $k \in \{0, \ldots, 9\}$ and the seed 99999. Each trained model had validation accuracy of around 81.63% on the clean dataset $S_{\mathrm{cl}}$, which did not change significantly when we retrained after adding poison samples and / or camouflage samples. Note that the efficacy of the camouflaged poisoning attack was more than 70% in most of the experiments. We provide a sample of the generated poisons and camouflages in Figure 9 in Appendix B.5.

### B.3.2 Support Vector Machines

| Attack type | Attack success | | Validation Accuracy | | |
|---|---|---|---|---|---|
| $(\varepsilon, b_p, b_c)$ | Poisoning | Camouflaging | Clean | Poisoned | Camouflaged |
| LF $(8, 0.2\%, 0.2\%)$ | 70% | 71.5% | 81.63 | 81.73 ($\pm$ 0.14) | 81.74 ($\pm$ 0.20) |
| LF $(16, 0.2\%, 0.2\%)$ | 100% | 40% | 81.63 | 81.64 ($\pm$0.03) | 81.6 ($\pm$0.02) |
| GM $(8, 0.2\%, 0.4\%)$ | 70% | 100% | 81.63 | 81.65 ($\pm$0.01) | 81.62 ($\pm$0.02) |
| GM $(16, 0.2\%, 0.4\%)$ | 100% | 70% | 81.63 | 81.65 ($\pm$0.03) | 81.63 ($\pm$ 0.02) |

Table 6: Camouflaged poisoning attack on linear SVM on Binary-CIFAR-10 dataset. The first column lists the threat model $(\varepsilon, b_p, b_c)$ and the camouflaging type "LF" for label flipping and "GM" for gradient matching.

In order to perform evaluations on SVM, we first convert the CIFAR-10 dataset into a binary classification dataset (which we term as Binary-CIFAR-10) by merging the 10 classes into two groups: `animal` ($y = +1$) and `machine` ($y = -1$)). Images (in both training and test datasets) that were originally labeled (*bird, cat, deer, dog, frog, horse*) are relabeled `animal`, and the remaining images, with original labels (*airplane, cars, ship, truck*), are labeled `machine`.

We train a linear SVM (no kernel was used) with the hinge loss. We evaluate both label flipping and gradient matching to generate camouflages, and different threat models $(\varepsilon, b_p, b_c)$; the results are reported in Table 6. Each poison and camouflage generation took about 40 - 50 seconds (for $b_p = b_c = 0.2\%$). Each trained model had validation accuracy of around 81.63% on the clean dataset $S_{\mathrm{cl}}$, which did not change significantly when we retrained after adding poison samples and/or camouflage samples. Note that the efficacy of the camouflaged poisoning attack was more than 70% in most of the experiments. We give examples of the generated poisons/camouflages in Appendix B.5.

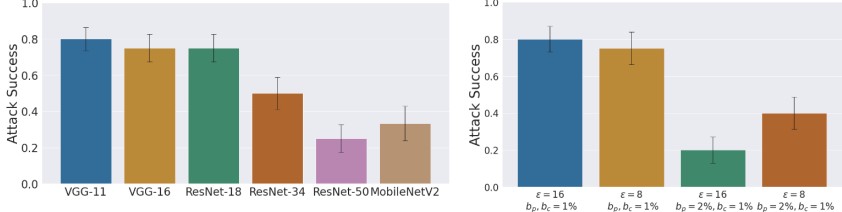

Figure 4: Efficacy of the proposed camouflaged poisoning attack on CIFAR-10 dataset. The left plot gives the success for the threat model $\varepsilon = 16, b_p = 0.6\%, b_c = 0.6\%$ for different neural network architectures. The right plot gives the success for ResNet-18 architecture for different threat models.

### B.3.3 Neural Network Experiments

**Implementation Details and Hyperparameters.** Each model is trained with cross-entropy loss $\ell(f(x, \theta), y) = -\log(\Pr(y = f(x, \theta)))$ on a single GPU using PyTorch [Paszke et al., 2019], and using mini-batch SGD with weight decay 5e-4, momentum 0.9, learning rate 0.01, batch size 100, and 40 epochs over the training dataset. Each training run took about 45 minutes. The poison and camouflage sets were generated using gradient matching by first defining the cosine-similarity

loss using `torch.autograd` and then minimizing it using Adam with a learning rate of 0.1. Each poison/camouflage generation took about 1.5 hours.[8]

**Further Results.**  We next elaborate on the results reported in Figure 3. In Table 7, we report the efficacy of the proposed camouflaged poisoning attack on different neural network architectures where the threat model is given by $\varepsilon = 16, b_p = 0.6\%, b_c = 0.6\%$. The reported results are an average over 5 seeds from 2000000000-2000001111. In the first column under attack success, we report the number of times poisoning was successful amongst the run trials, and in the second column, we report the number of times camouflaging was successful for the trials for which poisoning was successful.

In Table 8, we report the success of the proposed attack when we change the threat model, but fix the network architecture to be ResNet-18. Each experiment was repeated times 5 times with 8 restarts each time, and the mean success rate is reported. These experiments were conducted with 5 seeds from 2000011111-2111111111.

| Network Architecture | Attack success | | Validation Accuracy | | |
|---|---|---|---|---|---|
| | Poisoning | Camouflaging | Clean | Poisoned | Camouflaged |
| VGG-11 | 100% | 80% | 85.01 | 85.03 (± 0.37) | 85.10 (± 0.29) |
| VGG-16 | 80% | 75% | 87.68 | 87.42 (± 0.17) | 87.45 (± 0.26) |
| ResNet-18 | 80% | 75% | 82.13 | 81.88 (± 0.15) | 81.80 (± 0.12) |
| ResNet-34 | 80% | 50% | 82.45 | 82.61 (± 0.30) | 83.12 (± 0.93) |
| ResNet-50 | 80% | 25% | 81.02 | 81.76 (± 0.13) | 84.62 (± 0.71) |
| MobileNetV2 | 60% | 33% | 82.79 | 83.26 (± 0.25) | 85.47 (± 0.27) |

Table 7: Evaluating our proposed camouflaged poisoning attack on various model architectures on the CIFAR-10 dataset with the threat model $\varepsilon = 16, b_p = 0.6\%, b_c = 0.6\%$.

## B.4   Additional Details on Imagenette / Imagewoof Experiments

We evaluate the efficacy of our attack vector on the challenging multiclass classification problem on the Imagenette and Imagewoof datasets [Howard, 2019]. Imagenette is a subset of 10 classes (*Tench, English springer, Cassette player, Chain saw, Building/church, French horn, Truck, Gas pump, Golf ball, Parachute*) from the Imagenet dataset [Russakovsky et al., 2015]. The Imagenette dataset consists of around 900 images of various sizes for each class. In total, we have 13394 images which are divided into a training dataset of size 9469 and test dataset of size 3925. To perform training, all images are resized and centrally cropped down to $224 \times 224$ pixels.

Imagewoof [Howard, 2019] is another subset of Imagenet dataset consisting of 10 classes (*Shih-Tzu, Rodesian Ridgeback, Beagle, English Foxhound, Border Terrier, Australian Terrier, Golden Retriever,*

---

[8]Our implementations of Algorithm 1 and 2 are not optimized for computational efficiency, and the provided wall clock times simply serve as a proof of concept that our attack is practically implementable. All computations are performed on a CPU, but can be made significantly faster by using GPU.

| Threat model | | | Attack success | | Validation Accuracy | | |
|---|---|---|---|---|---|---|---|
| $\varepsilon$ | $b_p$ | $b_c$ | Poisoning | Camouflaging | Clean | Poisoned | Camouflaged |
| 16 | 1% | 1% | 100% | 80% | 82.13 | 81.98 (± 0.16) | 82.12 (± 0.21) |
| 8 | 1% | 1% | 80% | 75% | 82.13 | 82.21 (± 0.21) | 82.09 (± 0.23) |
| 16 | 2% | 1% | 100% | 20% | 82.13 | 82.31 (± 0.26) | 82.19 (± 0.24) |
| 8 | 2% | 1% | 100% | 40% | 82.13 | 82.43 (± 0.30) | 82.34 (± 0.27) |

Table 8: Evaluating our proposed camouflaged poisoning attack on various threat models with CIFAR-10 dataset trained on ResNet-18.

*Old English Sheep Dog, Samoyed, Dingo*). Imagewoof consists of around 900 images of various sizes for each class, and in total 12954 images which are divided into a training dataset of size 9025 and test dataset of size 3929. Similar to Imagenette, we resize all images and crop to the central $224 \times 224$ pixels before training.

We evaluate our camouflaged poisoning attack on two different neural network architectures-VGG-16 and ResNet-18, and different threat models $(\varepsilon, b_p, b_c)$ listed in Table 2 . Each model is trained on a single GPU with cross-entropy loss, that is minimized using SGD algorithm with weight decay 5e-4, momentum 0.9 and batch size 20. We start with a learning rate of 0.01, and exponentially decay it with $\gamma = 0.9$ after every epoch, for a total of 50 epochs over the training dataset. The poisons and camouflages were generated using gradient matching by first defining the cosine-similarity loss using `torch.autograd` and then optimizing it using Adam optimizer with learning rate 0.1.

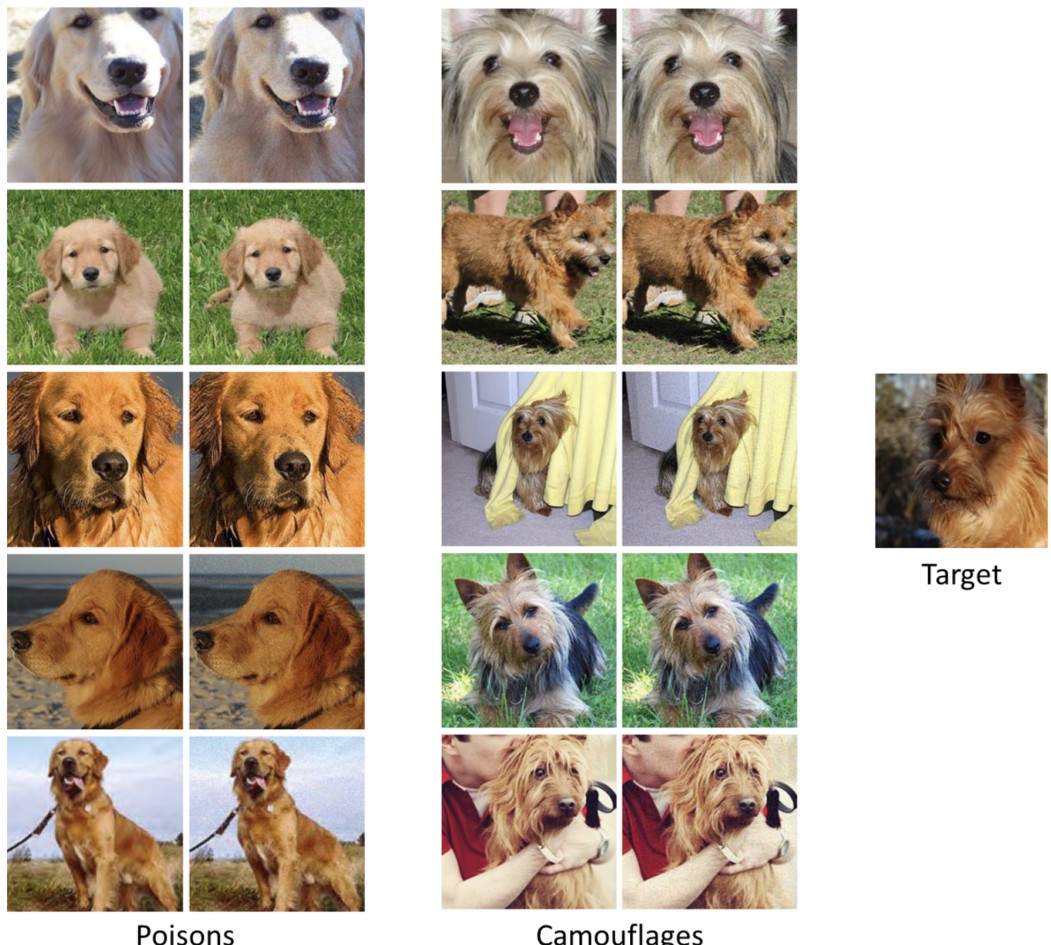

Poisons                    Camouflages                    Target

Figure 5: Visualization of poisons and camouflages on Imagewoof dataset. The first and the third columns shows the original images, and the second and the fourth columns shows the corrupted images (with added $\Delta$). The shown images were generated for a camouflaged poisoning attack on ResNet-18, with Seed = 2111111110, $b_p = b_c = 4.2\%$, $\varepsilon = 16$. The target and camouflage class is *Austrailian Terrier*, and the poison class is *Golden Retriever*.

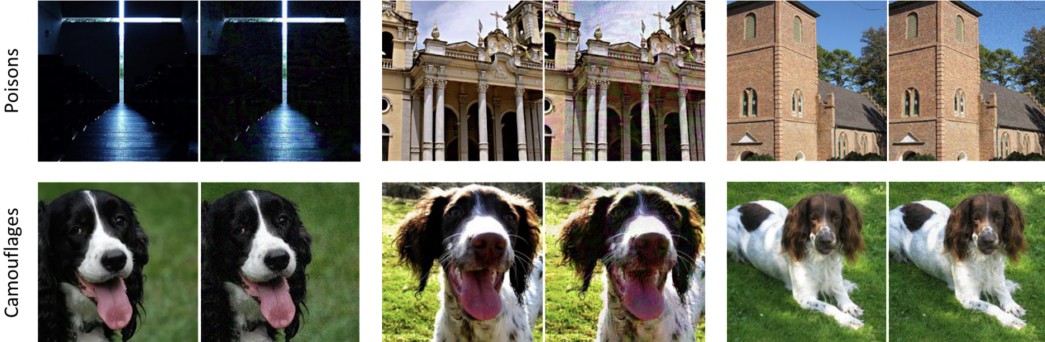

Figure 6: Some representative poison and camouflage images for attack on Imagewoof dataset. In each pair, the left figure is the original picture from the training dataset and the right figure has been adversarially manipulated by adding $\Delta$. The shown images were generated for a camouflaged poisoning attack on Resnet-18, with Seed = 10000005, $b_p = b_c = 6.6\%$ and $\varepsilon = 16$. The target and camouflage class is *English Springer*, and the poison class is *Building (Church)*.

## B.5 Visualizations

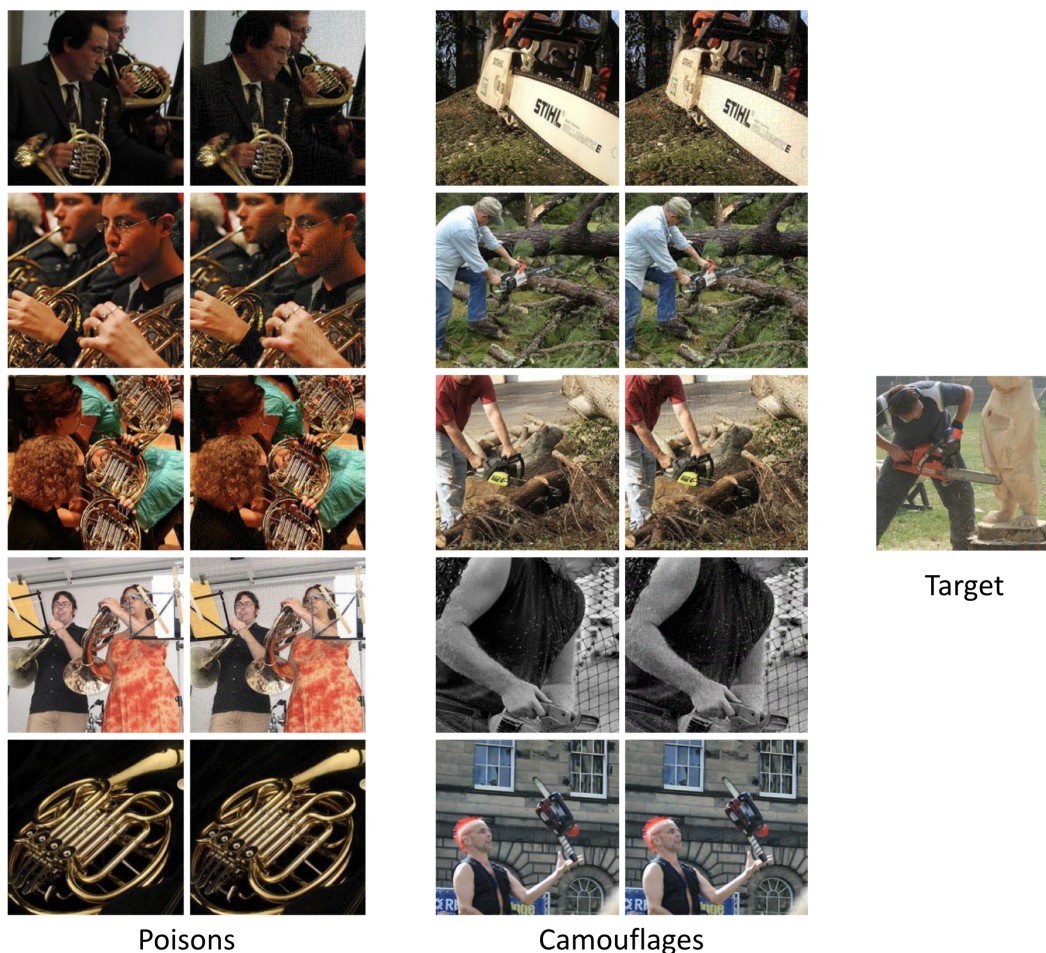

Figure 7: Visualization of poisons and camouflages on Imagenette dataset. The first and the third columns shows the original images, and the second and the fourth columns shows the corrupted images (with added $\Delta$). The shown images were generated for a camouflaged poisoning attack on ResNet-18, with Seed = 2000011111 and $\varepsilon = 8$. The target and camouflage class is *chain saw*, and the poison class is *French horn*.

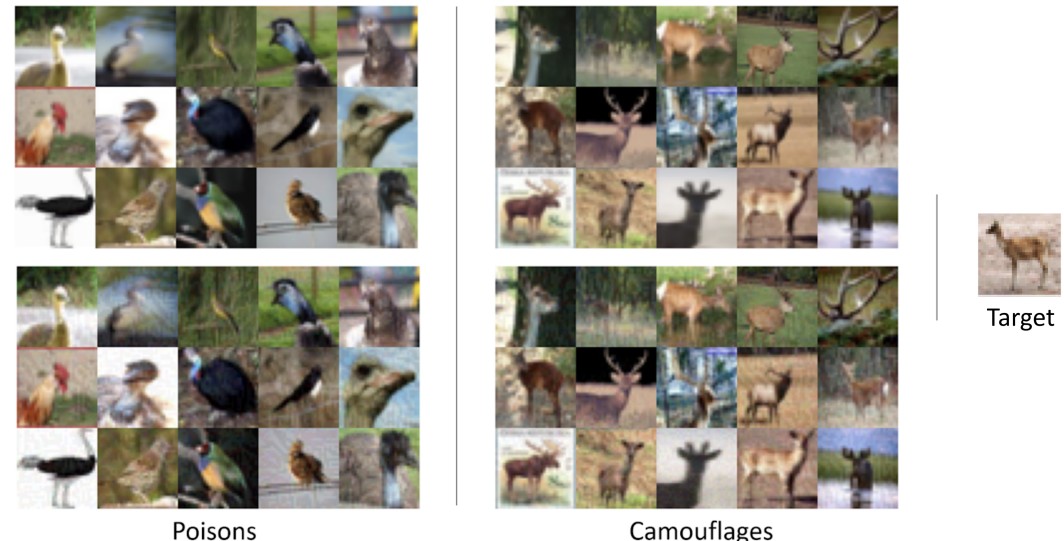

Poisons           Camouflages           Target

Figure 8: Visualization of poisons and camouflages on CIFAR-10 dataset (multiclass classification task). The top row shows the original images and the bottom row shows the corresponding poisoned / camouflaged images (with the added $\Delta$). The shown images were generated for a camouflaged poisoning attack on ResNet-18, with Seed = 2000000000, $\varepsilon = 8$, $b_p = 0.2, b_c = 0.4$, poison class *bird*, target class *deer*, and the target ID 9621.

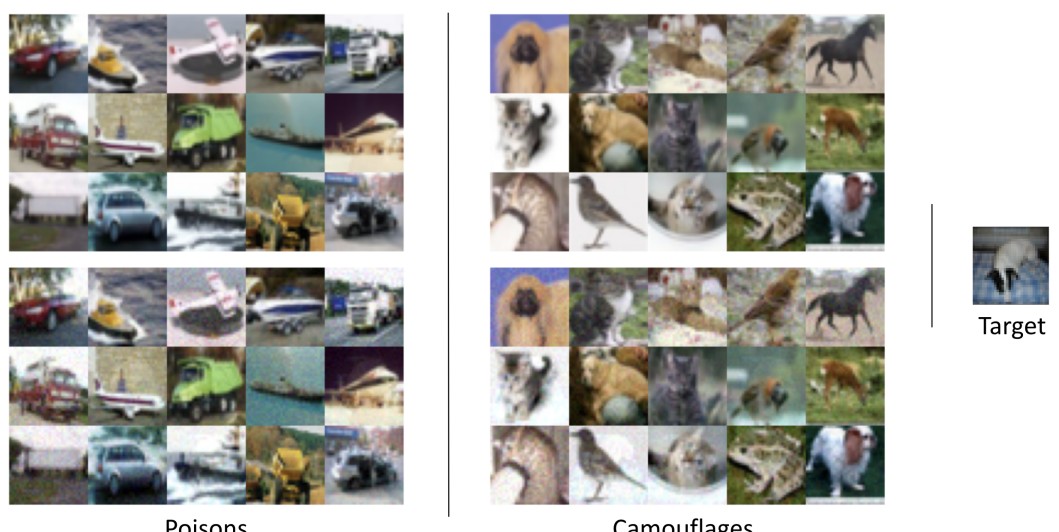

Poisons           Camouflages           Target

Figure 9: Visualization of poisons and camouflages on Binary-CIFAR-10 dataset (*animal* vs *machine* classification). The top row shows the original images and the bottom row shows the corresponding poisoned / camouflaged images (with the added $\Delta$). The shown images were generated for a camouflaged poisoning attack on SVM, with Seed = 555555, $\varepsilon = 16$, $b_p = 0.2, b_c = 0.4$, target ID 6646.

## B.6 Additional Experiments

### B.6.1 Ablation Study for Different Values of $b_p$ and $b_c$

In this section, we explore the effect of changing the relative sizes of the poison samples and camouflage samples. The experiments are performed on ResNet-18 for CIFAR-10 dataset with $b_p = 1\%$ and $b_c = \{0.5, 1, 1.5\}\%$ respectively, and vice-versa. We report the results in Table 9.

One may also wonder what is the price that an attacker has to pay in terms of the success rate of a poisoning (only) attack if it chooses to devote a part of the budget for camouflages. In Table 10, we report the results for comparing the success rate 1% poisons and 1% camouflages, to success rate with 2% poisons (and no camouflages.)

| Problem parameters | | Attack success | | Validation Accuracy | | |
|---|---|---|---|---|---|---|
| $b_p$ | $b_c$ | Poisoning | Camouflaging | Clean | Poisoned | Camouflaged |
| 1% | 1.5% | 83% | 80% | 0.822 | 0.8191 | 0.8264 |
| 1% | 0.5% | 83% | 40% | 0.845 | 0.8395 | 0.8432 |
| 1% | 1% | 83% | 80% | 0.822 | 0.8274 | 0.8244 |
| 0.5% | 1% | 33% | 50% | 0.822 | 0.8261 | 0.8194 |
| 1.5% | 1% | 83% | 20% | 0.822 | 0.8156 | 0.8118 |

Table 9: Effect of different sizes of poison and camouflage datasets on the success of the proposed camouflaged poisoning attack on CIFAR-10 dataset trained on ResNet-18, with $\varepsilon = 16$. The reported success rates are averaged over six different trials with seeds 200000111-211111111. For each experiment, we do 4 restarts for every poison and camouflage generation.

| Problem parameters | | Attack success | | Validation Accuracy | | |
|---|---|---|---|---|---|---|
| $b_p$ | $b_c$ | Poisoning | Camouflaging | Clean | Poisoned | Camouflaged |
| 0.5% | 0.5% | 33% | 50% | 0.822 | 0.8212 | 0.8238 |
| 1% | 1% | 83% | 80% | 0.822 | 0.8274 | 0.8244 |
| 2% | 0 | 83% | N/A | 0.822 | 0.817 | N/A |

Table 10: Comparison of success rate when the allocated budget of 2% is split as: (a) 1% poisons and 1% camouflage, and (b) 2% poison samples and 0% camouflages. The experiment is performed on CIFAR-10 dataset trained on ResNet-18 with $\varepsilon = 16$. The reported success rates are averaged over six different trials with seeds 200000111 - 211111111. For each experiment, we do 4 restarts per poison or camouflage generation.

### B.6.2 Robustness of the Proposed Attack to Random Deletions

We next explore the effect of random removal of the generated poison and camouflage samples on the success of our attack. In a data-scraping scenario, a victim may not scrape all the data points modified by an attacker. In Table 11, we report the effect on attack success when different amounts of the generated poison and camouflage samples are deleted uniformly at random.

| Amount removed | | Attack success | | Validation Accuracy | | |
|---|---|---|---|---|---|---|
| Poisons | Camouflages | Poisoning | Camouflaging | Clean | Poisoned | Camouflaged |
| 5% | 5% | 80% | 75% | 0.822 | 0.823 | 0.819 |
| 10% | 10% | 60% | 33% | 0.822 | 0.817 | 0.825 |
| 20% | 20% | 80% | 100% | 0.822 | 0.826 | 0.820 |
| 30% | 30% | 100% | 80% | 0.822 | 0.8215 | 0.818 |
| 40% | 40% | 20% | 100% | 0.822 | 0.825 | 0.817 |

Table 11: Effect of random removal of the generated poison and camouflage samples on the success of the proposed camouflaged poisoning attack on CIFAR-10 dataset trained on ResNet-18, with $\varepsilon = 16$. The reported success rates are averaged over 5 different trials with seeds 200001111 - 211111111. For each experiment, we do 4 restarts for every poison and camouflage generation.

### B.6.3 Transfer Experiments

In this section, we show that the poison and camouflage samples generated by the proposed approach transfer across models. Thus, an attacker can successfully execute the camouflaged poisoning attack, even if the victim trains a different model than the one on which the poison and camouflage samples were generated. We show the transfer success in Figure 10. The brewing network denotes the network architecture on which poison and camouflage samples were generated (we adopt the same notation as Geiping et al. [2021]). The victim network denotes the model architecture used by the victim for training on the manipulated dataset.

We ran a total of 3 experiments per (brewing model, victim model) pair using the seeds 2000000000-2000000011. Each reported number denotes the fraction of times when both poisoning and camouflaging were successful in the transfer experiment, and thus the attack could take place.

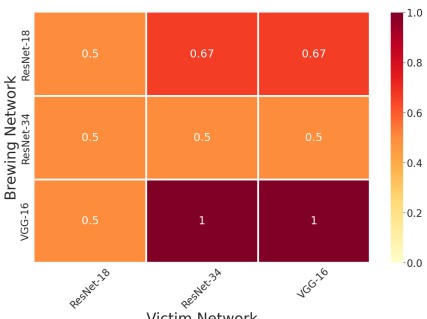

Figure 10: Transfer experiments on CIFAR10 dataset.

### B.6.4 Robustness to Data Augmentation

Data augmentation is commonly used to avoid overfitting in deep neural networks. In order to be applicable in the real life, our poisoning and camouflaging attacks must be successful even when the model is trained with data augmentation. In order to validate this, we evaluate our approach on CIFAR-10 dataset trained with data augmentation on ResNet-18 in the threat model $\varepsilon = 16, b_p = b_c = 1\%$; the results are in Table 12. The considered data augmentations are:

1. *No Augmentation*: Exact images from the training dataset are used.

2. *Augmentation Set 1*: $50\%$ chance that the image will be horizontally flipped, but no rotations.

3. *Augmentation Set 2*: $50\%$ chance that the image will be horizontally flipped, and random rotations in Uniform$(-10, 10)$ degrees.

The reported results in Table 12 are an average over 5 random seeds from "kkkkkk" where $1 \leq k \leq 5$. As expected, the validation accuracy for the model trained on clean dataset increased from 82%

percent when trained without augmentation, to 86% for augmentation Set 1 and 88% for augmentation set 2. The addition of data augmentation during training and re-training stages make it harder for poisoning to succeed and at the same time makes it easier for camouflaging to succeed.

| Data Augmentation | Attack success | | Validation Accuracy | | |
|---|---|---|---|---|---|
| | Poisoning | Camouflaging | Clean | Poisoned | Camouflaged |
| No Augmentation | 100% | 20% | 82% | 82% | 82% |
| Augmentation Set 1 | 86% | 33% | 86% | 85% | 86% |
| Augmentation Set 2 | 60% | 100% | 88% | 86 | 86% |

Table 12: Effect of data augmentation on our proposed camouflaged poisoning attack.

### B.6.5 Similarity of Feature Space Distance

A natural approach to defend against dataset manipulation attacks is to try to identify the modified images, and then remove them from the training dataset (i.e., *data sanitization*), e.g. Diakonikolas et al. [2019a]. For instance, one could cluster images based on their distance from their class mean image, or from the target image. This type of defense could potentially thwart watermarking poisoning attacks such as Poison Frogs [Shafahi et al., 2018]. As we show in Figure 11, such a defense would not be effective against our proposed poison and camouflage generation procedures, as the data distribution for the poison set and the camouflage set is similar to that of the clean images from the respective classes.

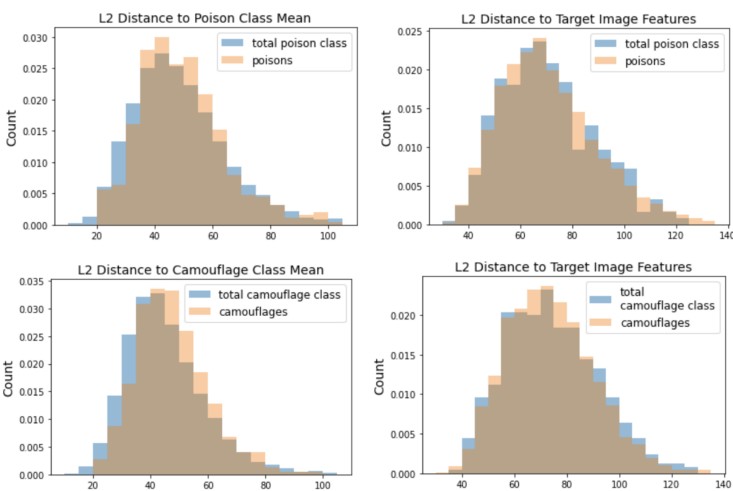

Figure 11: Feature space distance for our generated poison and camouflage set. The reported data was collected by a successful camouflaged poisoning attack on Resnet-18 model trailed on CIFAR-10 with seed 2000000000, $\varepsilon = 16$ and $b_p = b_c = 1\%$.

### B.6.6 Approximate Unlearning

In the experiments reported so far, we assumed that the victim performs exact unlearning, i.e. retrains the model from scratch on the leftover training data every time there is an unlearning request. However, in many cases, retraining from scratch for every unlearning request could be computationally expensive. This has inspired a plethora of research over the past few years in machine unlearning to develop approximate unlearning methods that are computationally faster than retraining from scratch [Sekhari et al., 2021, Guo et al., 2020, Bourtoule et al., 2021, Brophy and Lowd, 2021, Graves et al., 2021]. One may wonder if our attack also succeeds when performing approximate unlearning. Unfortunately, the algorithms dicussed in the prior works are infeasible for our large-scale deep learning setting; they either require strong structural properties (e.g., convexity [Sekhari et al.,

2021, Guo et al., 2020]) or require access to large memory [Bourtoule et al., 2021, Brophy and Lowd, 2021, Graves et al., 2021], etc. Hence, we were unable to evaluable most of these methods. However, we were successful in implementing the approximate unlearning methods in Graves et al. [2021] called *Amnesiac Unlearning* withing our available resources and report the results in Table 13.

To give the high level idea of *Amnesiac Unlearning*, note that the final model produced by multi-epoch mini-batch SGD like algorithms is given by:

$$\theta_{final} = \theta_{\text{initial}} + \sum_{e=1}^{E} \sum_{b=1}^{B} \Delta_{\theta_{e,b}}$$

where $\Delta_{\theta_{e,b}}$ is the incremental update using batch $b$ in epoch $e$. In *Amnesiac Unlearning*, the learner keeps track of which batches contain which sample points. When given an unlearning request, the learner computes the bathces $SB$ that contain the sensitive data (that requested for removal) and updates the model as

$$\theta_{\text{updated}} = \theta_{\text{initial}} + \sum_{e=1}^{E} \sum_{b=1}^{B} \Delta_{\theta_{e,b}} - \sum_{sb=1}^{S} B \Delta_{\theta_{sb}} = \theta_{\text{initial}} - \sum_{sb=1}^{S} B \Delta_{\theta_{sb}}.$$

The above update is followed by a few epochs of fine tuning on left over data samples to recover any loss in validation error. As reported in Graves et al. [2021], the above update method is particularly effective when the sensitive data comprises of about 1-2% of the training dataset, which is the case in our experiments. We refer the reader to Graves et al. [2021] for more details. The reported results in Table 13 were obtained by using the open source implementation of *Amnesiac Unlearning* by Graves et al. [2021]. We note that each experiment on CIFAR-10 took about 200GB of memory.

| Threat model | | | Attack success | | Validation Accuracy | | |
|---|---|---|---|---|---|---|---|
| $\varepsilon$ | $b_p$ | $b_c$ | Poisoning | Camouflaging | Clean | Poisoned | Camouflaged |
| 16 | 0.5% | 1% | 40% | 100% | 0.913 | 0.765 ($\pm 0.047$) | 0.722 ($\pm 0.060$) |
| 16 | 1% | 1% | 40% | 100% | 0.913 | 0.760 ($\pm 0.046$) | 0.765 ($\pm 0.046$) |

Table 13: Evaluating the success of our proposed camouflaged poisoning attack against *Amnesiac Unlearning* (an approximate unlearning technique) provided by Graves et al. [2021] using five randomly selected seeds provided in the table above. We remark that our attack continues to be effective even against approximate unlearning.

## B.7 Multiple Targets Scenarios

In addition, we performed a limited number of experiments on poisoning and camouflaging multiple targets using the gradient matching method. The results (as shown in Table 14) are comparable to the experiments performed by [Geiping et al., 2021] on multiple targets. We observe that our attack continues to be effective under multiple target settings. Furthermore, choosing a method that is more effective under the multi-target setting may increase the success of our attack.

| Threat model | | | Targets | | | Attack Success | |
|---|---|---|---|---|---|---|---|
| $\varepsilon$ | $b_p$ | $b_c$ | Targets | $b_p$ / Targets | $b_c$ / Targets | Poisoning | Camouflaging |
| 16 | 1% | 1% | 2 | 0.5% | 0.5% | 70% | 40% |
| 16 | 1% | 1% | 5 | 0.25% | 0.25% | 36% | 33% |
| 16 | 1% | 1% | 10 | 0.1% | 0.1% | 21% | 15% |

Table 14: Evaluating the success of our proposed camouflaged poisoning attack against multiple targets using the *Gradient Matching technique* on the CIFAR-10 dataset with default parameters.

## C  Alternative Approaches to Generate Poisons and Camouflages

Thus far, our experiments have focused solely on generating poison and camouflage points using the gradient matching technique. In this section, we show that our attack framework is robust to the

poison generation method: we generate a camouflaged poisoning attack using Bullseye Polytope poisoning—a clean-label targeted attack proposed by Aghakhani et al. [2021]. We believe that similar results should hold for a variety of other data poisoning techniques.

Bullseye Polytope (BP) maximizes the similarity between representations of the poisons and target. In doing so, it implicitly minimizes the alignment between poison and target gradients with respect to the penultimate layer, which captures most of the gradient norm variation [Katharopoulos and Fleuret, 2019]. In the transfer learning scenario, the poisons are crafted to have a similar representation to that of the target. Here, a linear layer is trained on the poisoned data using the representations obtained from a pre-trained clean model. The gradient of the linear model is proportional to the representations learned by the pre-trained model. Therefore, by maximizing the similarity between the representations of the poisons and the adversarially labeled target, the attack indeed increases the alignment between their gradients [Yang et al., 2022].

In our experiments, a poison set and a camouflage set were crafted and evaluated in a white-box setting, where the method has the highest average attack success rate [Schwarzschild et al., 2021]. Two pre-trained ResNet-18 models are used as the feature extractor to generate adversarial examples and as the victim, respectively. We set an $\ell_\infty$ perturbation budget of $\varepsilon = 7$ and perform BP for 1,000 iterations each to obtain poisons and camouflages. We then use Adam with a learning rate of 0.1 to fine-tune the victim's linear classifier on the poisoned dataset for 60 epochs. Finally, we re-load the victim model, repeat the fine-tuning process with the poisoned + camouflaged dataset and compare the results.

| Threat model | | | Attack success | | Validation Accuracy | | |
|---|---|---|---|---|---|---|---|
| $\varepsilon$ | $b_p$ | $b_c$ | Poisoning | Camouflaging | Clean | Poisoned | Camouflaged |
| 8 | 5 | 5 | 90% | 67% | 0.870 | 0.8794 ($\pm$0.012) | 0.8737 ($\pm$0.021) |
| 8 | 5 | 10 | 90% | 78% | 0.870 | 0.8727 ($\pm$0.002) | 0.8716 ($\pm$0.002) |

Table 15: Evaluating the success of our proposed camouflaged poisoning attack with *Bullseye Polytope poisoning* (a clean-label targeted attack) provided by Aghakhani et al. [2021] on the CIFAR-10 dataset using ten randomly selected seeds provided in the table above. We observe that our attack continues to be effective with different targeted poisoning methods.

