# OpenReview forum: "Hidden Poison: Machine Unlearning Enables Camouflaged Poisoning Attacks"
_NeurIPS.cc/2023/Conference — NeurIPS 2023 poster_

### Official Review · Reviewer_dUYW · 2023-06-30

**Soundness:** 3 good
**Presentation:** 3 good
**Contribution:** 3 good
**Rating:** 4
**Confidence:** 4

**Summary:**

The paper proposes a novel attack in machine unlearning. There are two step in the attack process. In the first step, the poisoned data and the camouflaged data are fed into the initial model training, together with the benign data. The machine unlearning process would take out the camouflaged data and the retraining from scratch process will be on the poisoned data and the benign data. Experiments are done to demonstrate the proposed attack.

**Strengths:**

•	The authors propose a novel attack in machine unlearning.

•	Evaluations are done on the label flipping and gradient matching setting for camouflaged data generation, two types of models (SVM and neural networks will multiple different network structures), and three datasets.

•	The paper is well-written and easy to follow.

**Weaknesses:**

•	The paper makes impractical assumptions:

        o	In step 4, the unlearned model is by training the clean ones and poisoned ones. This reviewer fails to understand if the initial training can identify the poisoning effect and require the camouflaged data to mask the poisoning effect, the why the machine unlearning procedure does not offer such capability, especially the authors adopt the retraining from scratch strategy to perform machine unlearning?

•	The paper makes a few false claims:

        o	This reviewer does not understand why the authors call the proposed approach clean-label when they have a poisoned set generated very similar to adversarial examples: looks the same for human but will be misclassified by the model. Even if the attack does not change the label directly, gradient matching would still change the optimization of the objective. Otherwise, if the label or the intermediate computation results is not changed and only the input data is slightly modified, the so-called attack becomes adversarial training.

        o	The authors are making speculations which are not true. The authors keep emphasizing the proposed approach can be applied to approximate unlearning methods in the literature. But this reviewer does not buy this claim. For example, Bourtoule et al., 2021 would trace back to the intermediate models before the data to be removed is in use. This approach could fail the proposed attack.

        o	In machine unlearning approach in this paper, which is retraining from scratch, is also a cost-heavy way to do machine unlearning. Therefore, the comments on 288-290 on the strong assumptions or taking too much memory is not as convincing.

        o	Typo: line 324, the second “learning” should be “unlearning”

**Questions:**

See weakness.

**Limitations:**

Not applicable.

---

> ### Author Rebuttal · Authors · 2023-08-10
>
> Unfortunately, we were not fully able to parse your concerns. Please find the following response to the best of our interpretation. We would love to engage further with the reviewer if any of their concerns are not answered:
>
> > Retraining from scratch vs approximate unlearning
>
> In machine unlearning, in both theory and practice, the goal is to output a model that is indistinguishable from the model trained from scratch on the leftover training dataset. Since this is the gold standard which all approximate unlearning approaches aim to achieve, we simply perform (our main) experiments by unlearning by retraining from scratch. However, in Appendix C.6.6 we also explore the success of our attack against an approximate unlearning approach that is applicable for large scale deep learning settings.
>
> Unfortunately, we are a small resource-constrained group, and perfoming experiments against the unlearning approach of Bourtoule et al., 2021 is not feasible for us: their method necessitates impractically large memory overheads, and at the cost of significant reduction of accuracy. Finally, regarding the reviewers concerns that retraining from scratch is cost-heavy, we remark that our poison and camouflage generation approach is actually quite efficient. We were able to perform each of our CIFAR-10 experiments in about 2 hours on a single GPU (when our code was not even GPU optimized) and we did not require extensive storage and memory (just need to store a trained model). This is much less than the extreme cost of Bourtoule et al, which necessitates storing multiple (on the order of 50 to 100) trained models.
>
>
> > This reviewer does not understand why the authors call the proposed approach clean-label when they have a poisoned set generated very similar to adversarial examples: looks the same for human but will be misclassified by the model
>
> An attack is called clean-label attack if visually looking at the image and the label pair, a human cannot tell if some noise is added or not. In particular, a picture that has a label dog should visually look like dog. All our attacks are clean label attacks, as is evident from the process of poisons and camouflages generation. This is described in more detail in the paper of Geiping et al.

---

> > ### Comment · Reviewer_oSYx · 2023-08-17
> >
> > As someone who has followed this field, I agree with the description by the authors of this attack as "clean label." Clean label attacks are those where the ground truth label of the image is not modified, as the authors describe here. For example, one of the first works to use the term clean label is Shafahi et al. 2018: https://arxiv.org/abs/1804.00792 It states in its abstract "The proposed attacks use 'clean-labels'; they don't require the attacker to have any control over the labeling of training data" In the case of this paper, the authors assert this is a clean label attack since an independent labeler looking at their images would label them as the "original" ground truth of the adversarial example.

---

> > > ### Author Response · Authors · 2023-08-17
> > >
> > > Thank you, we appreciate Reviewer oSYx's confirmation on the definition of clean label.
> > >
> > > In the final version, we'll include a more precise description of what the term entails, as well an attribution to the work of Shafahi et al., 2018, as we too believe this is the first usage of this term.

---

> > > > ### Author Response · Authors · 2023-08-19
> > > > **Follow up!**
> > > >
> > > > Since we are approaching the end of the rebuttal period, please let us know if you have any further questions or concerns. We would love to engage with the reviewer if the reviewer has any pending concerns.
> > > >
> > > > Thank you,
> > > > Authors

---

### Official Review · Reviewer_zrPH · 2023-07-03

**Soundness:** 3 good
**Presentation:** 3 good
**Contribution:** 3 good
**Rating:** 6
**Confidence:** 5

**Summary:**

This paper studies poisoning attacks against machine unlearning systems, where the attacker injects a set of poisoning samples and a set of camouflage samples. When both sets of samples are present in the training set, the trained model performs similarly to the clean model but shows adversarial behaviors and misclassified a particular sample into a wrong label when the camouflage samples are deleted. Throurgh evaluations, the proposed attack can achieve reasonable poisoning success and the camouflaged samples also serve well in preserving correct predictions when present.

**Strengths:**

1. The idea of attacking machine unlearning systems is very interesting.
2. The proposed attack, although majorly leverages the existing targeted poisoning attack, performs relatively effectively.
3. Evaluations show that the proposed attack may transfer across different model architectures and also unlearning methods.

**Weaknesses:**

1. The extension to the practically relevant setting of misclassifying multiple test points is not demonstrated.
2. The attack requires a significant amount of poisoning points to work for more complex datasets such as ImageNet.

**Questions:**

Overall, I think the problem studied in this paper is very interesting and the presentation of this paper is also very clear. However, I have the following concerns:
1. Misclassifying a single test point might not be practically relevant, attackers might be interested in investing a significant amount of poisoning points to misclassify as many test samples as possible. Although the authors discussed about the extension is straightforward, for practical implementation, I believe this will not work well. The original GradMatch attack performs poorly when the target number of samples exceeds maybe 5 images, not to mention that the proposed attack already performs inferior to the GradMatch attack in the standard training setting. So, I believe the authors may either tune down the statement slightly or demonstrate it through experiments (this will be ideal).
2. The authors mentioned that testing with different random seeds can lead to drastically different effectiveness and in practice, it is hard to know the random seed used by the victim (I hope I understand the paper correctly that the attacker and victim are indeed using different random seeds. Otherwise, the experiments probably be needed to rerun). In this regard, how can the attacker pick a good random seed to work well against the unknown seed by the victim? Repeating multiple times may not help as it is related to the unknown seed used by the victim.
3. The attack performs relatively poorly on more complex ImageNet datasets, and the performance of the GradMatch attack in the standard training is not listed. By checking the original paper, it seems the attack success of GradMatch is > 80% at <1% poisoning ratio, which means there is a big performance gap between the original GradMatch and the GradMatch in unlearning settings. This is probably a more worrying observation for me-- if there is an inherent barrier to poisoning complex models in unlearning settings, then we may have to worry too much about the poisoning threat. I guess this issue cannot be solved in a short amount of time but would be ideal if the authors can provide the attack performance when the total poisoning ratio is set as the common ratio used in prior poisoning literature (e.g., 1%). Related to this, the poisoning budget is usually 2x the budget of the poisoning standard setting, while in practice, attackers only have a limited budget. This implies the poisoning threat in machine unlearning will be less severe than poisoning in standard training.
4. There is an observation that the attack effectiveness varies drastically across model architectures. Do the authors have explanations for this?
5. When generating the camouflaged points, what is the intuition behind using the poisoned model $\theta_{cp}$ to generate the points? Is this a possible reason that leads to inferior performance of the attack on more complex datasets?

====================================================

This is to acknowledge that I have read the authors rebuttals carefully and my concerns are addressed.

---

> ### Author Rebuttal · Authors · 2023-08-10
>
> We are heartened to hear that the reviewer finds our paper to be an interesting attack, which is effective, and that we performed a thorough investigation which shows its efficacy across a variety of settings. We believe such a contribution to be above the bar for NeurIPS. We hope to assuage any concerns with our responses to the reviewer's thoughtful questions.
>
> > Multiple target points
>
>
> Thank you for the feedback. Indeed, since our attack is based on the Gradient Matching attack of Geiping et al., we inherit any weaknesses of their method. Targeting multiple points, even in the easier data poisoning setting, remains a challenging question at the frontier of data poisoning research, and a stronger attack for these settings would immediately result in stronger attacks in our setting. We will soften the language in our paper to reflect this. Nonetheless, we performed experiments with two target points and 1% budget (500 images) where we found that the success rate is 70% for poisoning and 40% for camouflaging, respectively.
>
> We also tried the harder setting with five target samples (where, as the reviewer points out, Geiping et al.'s attack is not very successful), and found the efficacy to be much lower - 36% and 33%, respectively. Recall that, as pointed out in our paper and Geiping et al.'s, even a low attack efficacy is problematic, as the adversary may repeat the attack until a successful attack is discovered, and then deploy it.
>
>
> > The authors mentioned that testing with different random seeds can lead to drastically different effectiveness and in practice, it is hard to know the random seed used by the victim (I hope I understand the paper correctly that the attacker and victim are indeed using different random seeds. Otherwise, the experiments probably be needed to rerun). In this regard, how can the attacker pick a good random seed to work well against the unknown seed by the victim? Repeating multiple times may not help as it is related to the unknown seed used by the victim.
>
> We confirm that we do not assume the attacker and the victim use the same seed. Furthermore, the randomness in our experiments results primarily from the attacker's random seed. Given successful poison and camouflage samples, the attack remains effective robust to changes in the victim's random seed. In our experiments, we used both pre-trained models and model checkpoint (trained with a different seed than the attacker) as the victim and were able to obtain comparable results.
>
> > Bigger models and different poison/camouflage budgets
>
> Unfortunately, we are a small group without significant access to computational resources, and thus we do not have enough resources to evaluate the full ImageNet. We instead evaluated these attacks on more tractable versions of ImageNet, called Imagenette and Imagewoof, that we used for experiments with high resolution images. We found that even small poison budgets like 300-400 images accounted for a larger fraction of the dataset (hence, the reported percentages are larger).
>
> We next address reviewers concerns about experiments with different poison and camouflage budgets. We want to point out that we already explore this in Appendic C.6.1 (Table 9, 10). We demonstrate that the attack continues to hold across different budget ranges.
>
> > There is an observation that the attack effectiveness varies drastically across model architectures. Do the authors have explanations for this?
>
> The success of our attack is conditional on the success of gradient-matching technique of Geiping et al. for data poisoning. Our reported variability across different model architectures comes from the variability of gradient-matching across different model architecture, as reported in Figure 3 of Geiping et al. A more successful data poisoning attack (which is not the focus of our work) should immediately result in a more effective camouflage attack. Such attacks remain at the frontier of data poisoning research.
>
> > When generating the camouflaged points, what is the intuition behind using the poisoned model to generate the points?
>
> The goal of camouflage samples is to cancel the effect of poison samples, when both are taken together. Thus, in order to cancel the effect of poisons, we generate camouflages on models trained on clearn + poison samples so that gradient matching for camouflage generation has some information about the poison samples.

---

> > ### Comment · Reviewer_zrPH · 2023-08-13
> > **Thanks for the response**
> >
> > Thanks for the rebuttal. I think some of my concerns are addressed, while others are not likely to be solvable within a short amount of time and the academic resource budgets. I would encourage the authors to discuss the performance of attacking multiple points in the paper and also to make it clear in the paper that the attack performance variation results from the randomness on the attacker's side, not the victim's side. I wish there could be more insights on why the performance varies across architectures, but also agree that poisoning on large models is still complicated and would not penalize the authors for not addressing this clearly. I am raising my score accordingly.

---

> > > ### Author Response · Authors · 2023-08-14
> > >
> > > Thank you for your consideration and response! Yes, we certainly will do so in the revised version of the paper.

---

### Official Review · Reviewer_VjL7 · 2023-07-05

**Soundness:** 3 good
**Presentation:** 3 good
**Contribution:** 2 fair
**Rating:** 5
**Confidence:** 3

**Summary:**

This paper proposes a new attack for security risks in the field of machine unlearning, namely the camouflaged data poisoning attack. The attacker adds poisoned and camouflaged points to the training data, and then triggers a deletion request to remove the camouflaged points, causing the model to misclassify the target points. The authors conduct an extensive evaluation to assess the performance of the proposed attack, and the results are promising.

**Strengths:**

- Popular topics
- Well-structured and easy to follow

**Weaknesses:**

- Unrealistic assumptions
- Needs more detailed explanations
- Needs more complex datasets

**Questions:**

The authors introduce a new attack surface in the context of machine unlearning called camouflaged data poisoning attacks. I like this topic because both data poisoning attacks and machine unlearning are two popular research topics. In addition, the authors clearly describe the threat model, methodology, and evaluation. I appreciate that.

However, I do have the following concerns.

- The authors design two algorithms for generating camouflage samples. The first one is through label flipping and the second one is through gradient matching. My concern is that the second scenario where an attacker has access to model gradients is unrealistic, especially in so-called image-based social media platforms. I do not believe that such a platform would have white-box access to the user. Although the authors discuss this point in the last section, I wish the authors would focus more on black-box settings.

- Figure 1 provides the attack performance of the newly designed poisoning attack. Here, I wonder why the attack success rate on ResNet-50 is much lower than other model architectures. I hope the authors will have more discussion on this. I conjecture that one of the reasons behind this may be the much larger model complexity of ResNet-50. Therefore, I would further like the authors to add an ablation study on the impact of target model complexity.

- Following the above, I am also curious about how the proposed attack performs on more complex datasets. Specifically, all experiments are conducted on 10-class datasets. However, in real-world scenarios, the task is likely to be more complex, with more than 10 classes. Therefore, I would like the authors to perform an investigation on more complex datasets, such as CIFAR-100 and ImageNet.

**Limitations:**

Insufficient experiments, conclusions may not generalize to complex architectures and datasets

---

> ### Author Rebuttal · Authors · 2023-08-10
>
> We appreciate that the reviewer likes the direction, and for their thoughtful questions to keep us honest! We have investigated all these questions, and found that the attack is fairly robust to the modifications suggested. Details follow.
>
> > Extension of the attack to the black-box setting.
>
> Thanks for this interesting concern. Our existing transfer experiments in Appendix C.6.3 address this concern, where we show that the poisons and camouflages generated on model A can be used to attack a model B. Thus, the attacker does not need gradient access to model B, and can generate the poisons/camouflages "in-house" on model A.
>
> Exploring this further for more black-box setting, where the attacker does not have gradient access to any model is an interesting future research direction. We believe that techniques explored in Geiping et. al. 2021 (section 5.2) for black-box gradient matching attack should be directly extendable to implement our attack in a black-box setting.
>
> Finally, we also want to point out to the reviewer that we have additional experiments done using the Bullseye Polytope method in Appendix D which focuses on (more black-box) transfer learning scenarios.
>
> If the reviewer remains skeptical, we're happy to run additional experiments to assuage any concerns.
>
> > Figure 1 provides the attack performance of the newly designed poisoning attack. Here, I wonder why the attack success rate on ResNet-50 is much lower than other model architectures. I hope the authors will have more discussion on this. I conjecture that one of the reasons behind this may be the much larger model complexity of ResNet-50. Therefore, I would further like the authors to add an ablation study on the impact of target model complexity.
>
> The success of our attack is conditional on the success of gradient-matching technique of Geiping et al. for data poisoning. In their work, as reported in Figure 3, the attack efficacy is smaller in Resnet-50 as compated to Resnet-18. Our reported numbers follow a similar trend and range. Given a more successful data poisoning attack on Resnet-50 (not the focus of our work), we are confident that it would translate to a more effective attack in our setting.
>
> Regarding an ablation study regarding different models, please find this experiment in Section C.3.3 (Table 7).
>
>
> > Following the above, I am also curious about how the proposed attack performs on more complex datasets. Specifically, all experiments are conducted on 10-class datasets. However, in real-world scenarios, the task is likely to be more complex, with more than 10 classes. Therefore, I would like the authors to perform an investigation on more complex datasets, such as CIFAR-100 and ImageNet.
>
> We followed the reviewer's suggestion and performed additional experiments on CIFAR-100. We found that our attack is still effective. In particular, the poison success rate is 60% (80% if we consider top 5), and the camouflage success rate is 100%, with $b_p$ = 200,  $b_c$ = 400 and $\epsilon$ = 16. We will add details of this experiment to the final paper.
>
>
> Unfortunately, our group does not have access to the computational resources required to run experiments on ImageNet.

---

> > ### Author Response · Authors · 2023-08-19
> > **Follow up!**
> >
> > Since we are approaching the end of the rebuttal period, please let us know if you have any further questions or concerns. We would love to engage with the reviewer if the reviewer has any pending concerns.
> >
> > Thank you,
> > Authors

---

> > > ### Comment · Reviewer_VjL7 · 2023-08-19
> > >
> > > Thank you for your response!
> > >
> > > > Our existing transfer experiments in Appendix C.6.3 address this concern...
> > >
> > > The transferability study does alleviate some of my concerns regarding the black-box scenario. However, as I previously noted, the performance difference between ResNet-50 and ResNet-18 caught my attention. I understand that this might be influenced by the success of the poisoning attack, but it prompts me to question the conditions under which transferability truly comes into play.
> > >
> > > > performed additional experiments on CIFAR-100...
> > >
> > > I appreciate the effort put into testing the attack on CIFAR100. Nevertheless, since it's quite similar to CIFAR10, its significance might be limited. It would be more impactful if similar experiments could be conducted on a larger and more diverse dataset like ImageNet.
> > >
> > > > Unfortunately, our group does not have access to the computational resources required to run experiments on ImageNet.
> > >
> > > The point about computational resources caught my attention. If launching the attack requires substantial resources even for the authors, does this suggest that it's even more infeasible for potential adversaries?
> > >
> > > In conclusion, while there are still some lingering concerns - I do believe experiments on larger datasets like Imagenet are necessary - I appreciate the authors' clarification regarding the transferability. I am raising my score from 4 to 5.

---

> > > > ### Author Response · Authors · 2023-08-21
> > > > **Response to computational resources for Imagenet**
> > > >
> > > > Thank you for your feedback. We want to clarify our comment that **" our group does not have access to the computational resources required to run experiments on ImageNet"**. We have been running our experiments on Google Cloud and found it practically impossible for us to upload the entire ImageNet dataset for our experiments. Thus, we resorted to smaller Imagenette and Imagewoof datasets (10 classes instead of 1000 classes). In terms of computation, the time taken by the adversary to generate camouflages and poisons via gradient matching for ImageNet should be smaller than the time taken to train a model from scratch. Thus,  an adversary capable of training a model on Imagenet is likely to have the resources needed to implement the attack.

---

### Official Review · Reviewer_oSYx · 2023-07-29

**Soundness:** 4 excellent
**Presentation:** 4 excellent
**Contribution:** 4 excellent
**Rating:** 7
**Confidence:** 4

**Summary:**

This paper introduces a very interesting novel attack vector for tampering with the behavior of a machine learning model by influencing its training dataset. It poses that the practice of "unlearning" examples that are requested to be taken down can be exploited. The attack proceeds in several steps: 1. The adversary poisons the training data with two types of poisons: camouflages and regular poisons. 2. The adversary asks that the camouflages be unlearned (as might be their right under some legislation). 3. Unlearning is applied by the model developers, which triggers the poisoning effect and some examples are misclassified at inference time of the "unlearned" model.

**Strengths:**

This is a strong and novel attack.

**Weaknesses:**

I believe the biggest limitation of this work is that experiments are only conducted on small computer vision datasets. It would be very interesting to find out if these poisons work against modern text-to-image or large language models.

**Questions:**

No questions from me.

**Limitations:**

No concerns here.

---

> ### Author Rebuttal · Authors · 2023-08-10
>
> Thank you for your positive support for the paper! We agree that these other settings are also interesting. However, rather than cover many disparate settings poorly, we chose to investigate computer vision datasets in extreme depth, evidenced by our paper being already quite long (27 pages), with a broad range of ablations and investigations. We believe these additional settings mentioned are worth investigation in follow-up work.

---

### Official Review · Reviewer_eD7Z · 2023-08-01

**Soundness:** 3 good
**Presentation:** 2 fair
**Contribution:** 3 good
**Rating:** 6
**Confidence:** 4

**Summary:**

This paper introduces a clean-label targeted data poisoning attack for machine unlearning pipelines. Gradient matching [A] is used to generate the poisons, and experiments use the retraining from scratch method for machine unlearning.

The core idea is for an attacker to introduce poisons and camouflages to the clean dataset. The camouflages mask the effect of poisons, so the victim model trained on the combined dataset (clean + poison + camouflage) doesn't exhibit poisoned behavior. After this, the attacker makes an unlearning request to remove the camouflages. After the unlearning process, the model exhibits the poisoned behavior. The poisoning aims to misclassify a specific test sample to a chosen target class.

[A] Jonas Geiping, et al. “Witches' Brew: Industrial Scale Data Poisoning via Gradient Matching.” International Conference on Learning Representations. 2021.

**Strengths:**

The paper proposes an original attack idea – a targeted poisoning attack in a machine unlearning setup.

The optimization method used for generating the poisons (Gradient Matching) is a well-known method from poisoning literature to solve the bi-level problem. The experimental setup is decent with experiments covering a handful of datasets (CIFAR-10, ImageNette and Imagewoof) and machine learning models (SVMs, VGG, ResNets).

I was intrigued by the question the paper poses, “We thus identify a new technical question of broader interest to the data poisoning community: Can one nullify a data poisoning attack by only adding points?” I agree that this direction of study might be of significant interest to the community.

**Weaknesses:**

There are certain weaknesses in the claims of the paper, and the experiments are not thorough enough. They need improvement.

1. (Line 216) “camouflaging was successful ...  provided that poisoning was successful. A camouflaged poisoning attack is successful if both poisoning and camouflaging were successful.” This explanation doesn't seem consistent with the numbers reported in the tables. If the camouflaging success rate is conditional on the poisoning being successful, I would think camouflaging success rate ≤ poisoning success rate. But that doesn't seem to be true in Row 3 of Table 1. Furthermore, if both these numbers are independent of each other, the tables should also report when both happen, as mentioned in the last line of the quoted text.

2. (Footnote) “We diverge slightly from the threat model described above, in that the adversary modifies rather than
introduces new points. We do this for convenience and do not anticipate the results would qualitatively change.”
I do not think this is true. If we have a clean image and also the (same clean image + perturbation) in the poisoning dataset, wouldn't it change the results?

3. (Amount of training – Line 244) “Each model was trained to have validation accuracy between 81-87% (depending on the architecture)”.
81-87% on CIFAR-10 seems to be quite a low number. It should be easy to achieve 90-95% on CIFAR-10 with VGG-11, VGG-16, Resnet-18, and Resnet-34. These models seem to be under trained. Given that no experiments are performed on ImageNet and compute shouldn't be a bottleneck for CIFAR-10 training, the models need to be trained more so that the experimental results have more value for future research. See https://github.com/kuangliu/pytorch-cifar for reference numbers.

4. (Extension to a set of targets) The attack threat model used here aims to target a single image from the test set. Though this is a starting point, the applicability of such a poison is quite limited. A natural extension would be to do this for a collection of targets. Though this is mentioned as future work in the discussion section, I would recommend that being a part of this current work.

**Questions:**

There are some parts of the text which are not clear and will benefit with additional explanation.

1. (Line 172) What is $b_p$ and $b_c$?
2. (Line 173) What is $P$?
3. (Line 265) What is the validation accuracy for these models. I do not see them mentioned in the main text / appendix.

**Limitations:**

A discussion on potential negative societal impact is missing. As this is a poisoning attack paper with implications on real-world deployments of machine learning models, I would recommend the authors to include a comment.

---

> ### Author Rebuttal · Authors · 2023-08-10
>
> We thank the reviewer for their kind comments. They give strong support for the paper, pointing out novelty and interest of the attack, thoroughness of the investigation (covering several datasets and models) and the fact that we highlight new technical questions for the data poisoning community. We put significant effort into conveying these points, and believe they make our paper well above the NeurIPS bar. We have thoroughly tested the robustness of this vulnerability, evidenced by the length of our submission (27 pages) filled with numerous experiments, and through our code release, challenge the community to identify deficiencies in our attack. The reviewer identifies certain weaknesses, which we believe to be issues primarily with our writing rather than technical in nature, which we attempt to clarify. We also ran additional experiments to assuage the reviewer's concerns.
>
> > “camouflaging was successful ... provided that poisoning was successful.
>
> We always report conditional success. The reported number for the poison success column (e.g. in Table 1) denotes the percentage of trials for which the poison succeeded. The reported number for the Camouflaging success column denote the percentage of trials for which camouflaging successed amongst all the trials where poisoning succeeded. Since the reported numbers are percentages they could not be directly compared. Formally, they are defined as:
>
> $\text{poisoning success} = \frac{|\text{Number of trials for which poisoning succeeded}|}{|\text{Total number of trials}|}$
>
> and
>
> $\text{Camouflaging success} = \frac{|\text{Number of trials for which camouflaging succeeded}|}{|\text{Total number of trials for which poisoning succeeded}|}$.
>
> If the reviewer prefers, we are happy to additionally report the unconditional success rate.
>
> >“We diverge slightly from the threat model described above, in that the adversary modifies rather than introduces new points...”
>
> In theory, the model where the adversary can modify (rather than only add points) may be stronger, since it gives them the power to both remove and add points. However, we are unaware of any situation in the data poisoning literature where these two models qualitatively differ. In addition, we assume the adversary is given the points to attack uniformly at random, which further reduces their power and bringing the two adversary models closer together. As for the specific situation mentioned by the reviewer: note that the unmodified images are separate from the perturbed images in our threat model, so this instance could not arise.
>
> > "81-87% on CIFAR-10 seems to be quite a low number..."
>
> Thanks for pointing this out. The issue is that, in some of our initial experiments, we did not use data augmentation. However, Section C.6.4 and C.6.6 of the submission use data augmentation and achieve accuracy 88%-92%, which is more in line what the numbers you outline. They show that the attacks remains successful in these settings. We will re-run other experiments to include data augmentation. In particular, we already did this with multiple targets to address your next question: we achieved a validation accuracy of 91-92% on CIFAR-10, with an attack success rate of 50% for poisoning and 40% for camouflage, with .4% poisoning and camouflaging of the training set. We will refresh the numbers in the paper to reflect these experiments.
>
>
>
> > (Extension to a set of targets) The attack threat model used here aims to target a single image from the test set
>
> As mentioned in footnote 6, it is straightforward to extend our attack to multiple targets by changing the objective to a sum over losses on the target points. We demonstrated this with experiments on two targets on CIFAR-10 (including data augmentation) and found that our attack continues to be effective. In particular, the attack success rates is 50% for poisoning and 40% for camouflaging. Recall that our attack is based on Witches' Brew, which fails to design effective attacks against even as few as five targets, even for the easier poisoning setting without camouflage. Designing more effective attacks for multiple targets is at the frontier of research on data poisoning, and such attacks would immediately result in better attacks for our setting.
>
> We will add further discussions on all of the above mentioned comments, and well as additional experiments, to the final version of the paper.
>
>
>
> ## Answer to specific questions
> > What is $b_p$ and $b_c$?
>
> As mentioned in line 132, $b_p$ is the percentage of training sampled used as poisons, and $b_c$ are the perfectage of training samples used as camouflages.
>
> > (Line 173) What is P?
>
> As mentioned in line 148, P denotes the total number of poison samples. Technically, $P = b_p \times |S_{\mathrm{cl}}| / 100$.
>
>
> >(Line 265) What is the validation accuracy for these models. I do not see them mentioned in the main text / appendix.
>
> We apologize for missing out this important statistic. For Imagenette, the validation accuracy usually falls between 75%-80%. For imagewoof, the accuracy was lower at around 67%-72%. We will add a table in the final version of the paper.
>
> >Discussion on potential negative societal impact is missing
>
> Thanks for raising this concern. Indeed, any paper exposing a vulnerability has the potential for negative social impact. We'll add discussion in the final version.

---

> > ### Author Response · Authors · 2023-08-19
> > **Follow up!**
> >
> > Since we are approaching the end of the rebuttal period, please let us know if you have any further questions or concerns. We would love to engage with the reviewer if the reviewer has any pending concerns.
> >
> > Thank you,
> > Authors

---

> > > ### Comment · Reviewer_eD7Z · 2023-08-21
> > >
> > > Thank you for the detailed response.
> > >
> > > 1. It's now clear to me from the spelled out formulae of the metrics that the authors report conditional success of the camouflaging attack. It will be helpful to work these into the text either in words or as it is.
> > >
> > > Reviewer VjL7 raises a good point about low camouflaging performance for ResNet-50 (and MobileNetV2). Based on the metric definitions, a low poisoning success rate on ResNet-50 shouldn't influence camouflaging success rate. Even if one gets 25% poisoning rate, it's possible to get 100% camouflaging rate (Table 2, Row 1). Moreover, ResNet-50 gets 80% attack success rate, which is relatively high. (I am assuming Table 7 and Fig 4(left) report the same numbers.) It might be the case that on increasing model size, the attack efficacy decreases (as seen from the trend on ResNets (18, 34, 50)), which should be clarified.
> > >
> > > 2. I do not have a strong counterpoint to “However, we are unaware of any situation in the data poisoning literature where these two models qualitatively differ.” I'll need to think more about this, and it won't affect my final decision.
> > >
> > > 3. It will be great if the updated numbers with data augmentation training are added.
> > >
> > > 4. Would using a Bullseye Polytope objective improve camouflaging attack performance over Witches' Brew? As far as I know, it is designed to be effective for multiple targets.
> > >
> > > 5. Thank you for pointing out the locations where the symbols are defined.
> > >
> > > Additional Typo
> > >
> > > “Australian Terrier” (Fig 5 caption, Pg 20)

---

> > > > ### Author Response · Authors · 2023-08-21
> > > > **Thanks!**
> > > >
> > > > Thank you for your feedback.
> > > >
> > > > We will definitely add updated numbers with data augmentation training and clarify the definitions in the final version of the paper. After this is done, we should be able to confirm and discuss the effect of the model size on the attack success.
> > > >
> > > > **Incorporating Multiple Targets and comment about Bullseye Polytope"**
> > > >
> > > > From our understanding, the  Bullseye Polytope procedure of Aghakhani, et al. 2021 considers a multi-target mode that generates poisons by using multiple images of the same target object (e.g. from different angles, positions, etc). Note that this is different from generating poisons for multiple (and independent) targets, which seems to be what the reviewer is requesting. We agree that extending camouflaged poisoning attacks, or even vanilla data poisoning attacks, to work for multiple independent targets is an interesting research direction. However, this is beyond the scope of our paper, but if resources permit, we would be happy to add more experiments for multi-targets with  Bullseye Polytope in the final version of the paper.
> > > >
> > > > We also remark that we did run some experiments with Bullseye Polytope for generating our attack. We refer the reviewer to Appendix D for more details! We anticipate Bullseye Polytope to work better than gradient matching for multi-target settings since it generally requires way fewer poison images, e.g. 5-10 as compared to 200-500 for gradient matching.

---

> > > > ### Comment · Reviewer_eD7Z · 2023-08-21
> > > >
> > > > Thank you for the discussion. The authors tried their best to address my concerns. I am willing to raise my score.

---

### Decision · Program_Chairs · 2023-09-21

**Decision:**

Accept (poster)

**Comment:**

All reviewers except one (the last reviewer) are in favor of accepting this paper. The AC believes the rebuttal addresses most of the concerns raised by the last reviewer, e.g, clean label attack. Hence, the AC recommends accepting this paper.